# Advances in Nanotechnology for Cancer Immunoprevention and Immunotherapy: A Review

**DOI:** 10.3390/vaccines10101727

**Published:** 2022-10-16

**Authors:** Navami Prabhakar Koyande, Rupali Srivastava, Ananya Padmakumar, Aravind Kumar Rengan

**Affiliations:** Department of Biomedical Engineering, Indian Institute of Technology Hyderabad, Kandi, Sangareddy 502285, India

**Keywords:** cancer immunotherapy, cancer vaccines, immunomodulation, clinical trials, tumor antigens

## Abstract

One of the most effective cancer therapies, cancer immunotherapy has produced outstanding outcomes in the field of cancer treatment. However, the cost is excessive, which limits its applicability. A smart way to address this issue would be to apply the knowledge gained through immunotherapy to develop strategies for the immunoprevention of cancer. The use of cancer vaccines is one of the most popular methods of immunoprevention. This paper reviews the technologies and processes that support the advantages of cancer immunoprevention over traditional cancer immunotherapies. Nanoparticle drug delivery systems and nanoparticle-based nano-vaccines have been employed in the past for cancer immunotherapy. This paper outlines numerous immunoprevention strategies and how nanotechnology can be applied in immunoprevention. To comprehend the non-clinical and clinical evaluation of these cancer vaccines through clinical studies is essential for acceptance of the vaccines.

## 1. Introduction

The second leading cause of death worldwide is cancer, and this number is steadily increasing [1]. Although numerous therapeutic approaches have been used to improve the overall survival of cancer patients, the mortality rate is still high. These approaches include chemotherapy, radiation therapy, and surgical excision of the tumor. Although immunotherapy has emerged as a viable cancer treatment in recent decades, it continues to have several limitations. Therefore, there is no specific treatment for cancer [2,3,4]. Recent advances have been to prevent the modality from occurring, as its treatment is difficult. Chemoprevention and immunoprevention have surfaced as the two major strategies for the prevention of cancer. Chemoprevention has several disadvantages, like high toxicity, and so its application is limited. Immunoprevention, on the other hand, has lower side effects, low toxicity, ease of delivery and long term effects, due to the generation of immunological memory [5,6,7,8,9,10,11].

William Coley was the first to propose using the immune system to combat cancer. He treated recurring sarcoma with Coley’s toxins, which contained dead Streptococcus pyogenes and Serratia marcescens [12,13]. Paul Ehrlich later showed that the immune system can recognize and kill tumors [14]. The notion of immunosurveillance to destroy tumor cells was developed and it was suggested that the identification of neo-antigens on tumor cells can protect the body against tumor cells [15]. We now understand that the immune system can both suppress and promote tumor progression by immunoediting [16]. Hence, there is a possibility of modulating the immune system to prevent and treat tumors. However, applications of immune-based strategies require deep understanding of the immune system. Recently, nanoparticles have been extensively used in both immunopreventive and immunotherapeutic strategies to overcome the limitations of conventional immune-based strategies. Various kinds of nanoparticles have now been designed that are able to carry numerous types of anti-tumor components, like peptides, nucleic acids etc. These advances have significantly improved the status and outcomes of many immunotherapeutic and immunopreventive strategies. 

## 2. Role of Immune System in Cancer

This review discusses the role of the immune system in the development of cancer, as well as how to distinguish between premalignant lesions and malignant tumors. The use of monoclonal antibodies, immune checkpoint inhibition, cancer vaccines, and other immunotherapy methods are then described. In this way, a deeper understanding of immunotherapy’s present difficulties and how nanotechnology has assisted in resolving these problems is gained. Then, with the aid of recent preclinical and clinical investigations, we list numerous immunoprevention techniques and recent advances in nanotechnology for cancer immunoprevention. Subsequently, the review explores the details of anti-cancer vaccines and their potential applications in immunoprevention. 

Tumors are susceptible to both suppression and promotion by the immune system. For instance, the proinflammatory cytokine IL-6 controls the proliferation and survival of tumor cells by activating Notch-3 and upregulating the hypoxia response protein, carbonic anhydrase IX. This fosters the survival of breast cancer stem cells in hypoxic environments [17,18]. Dendritic cells (DCs), on the other hand, are antigen-presenting cells (APCs) that process and deliver antigens from tumor cells to the immune system [19]. Cytokines are also critical for anti-tumor response, as for T cells to exert their cytotoxic effects on tumor cells, other cytokines, including interferon (IFN)- γ and tumor necrosis factor (TNF)-α, must also be secreted [20]. T cells activation further secretes various cytokines and enzymes, like perforin-granzyme, that destroy malignant cells [21]. 

NK cells are another essential component in the removal of tumors [22]. On the contrary, components of the immune system, like tumor associated macrophages, regulatory T cells (Tregs) etc., promote the malignant state [23,24]. Tregs promote tumors by various mechanisms, like disabling T cell activation, secreting inhibitory cytokines, like IL-10, transforming growth factor (TGF)-β, interfering with the T cell function and upregulation of cytotoxic T-lymphocyte- associated protein (CTLA)-4 [25,26]. By producing cytokines, including TNF-α, IL-12, and IL-6, M1 macrophages have an anti-cancer effect, but M2 macrophages promote the proliferation of cancer cells by producing IL-10, TGF-α [27]. Last, but not least, immunological checkpoints, such as CTLA-4 and programmed cell death protein 1 (PD-1), are also involved in tumor progression. While CTLA-4 binds to CD28 ligands (CD80/CD86) to hinder T-cell priming, PD-1 attaches to its ligands (PD-L1/PD-L2) on tumor cells to induce T-cell death and inhibit the effector functions of cytotoxic T cells (Tc cells) [28,29,30,31]. All of these immune components enable tumor development or escape. Overall, we may conclude that altering the immune system to increase tumor-suppressing components and suppress tumor-promoting components could be useful for both treating and preventing tumors.

## 3. Distinguishing Premalignant from Malignant Lesions

Premalignant refers to a condition that has the potential or propensity to develop into cancer. These lesions raise a person’s chance of getting cancer because they are considered to be a hallmark of several malignancies. Knowing the difference between precancerous and cancerous lesions is crucial to choosing the right immune-based therapy. Precancerous lesions and cancerous lesions differ greatly from one another, making it simple to tell them apart (Table 1). The morphology of tumor cells is influenced by the site of cancer development, changes in physiological, cellular, and molecular characteristics, and the environment surrounding the tumor. The tumors continue to develop to an advanced stage and metastasize to an entirely new site [32]. In contrast, premalignant lesions are restricted to the place of origin. For instance, ducts or the tissues’ epithelial layers are still the only places where epithelial malignancies can occur [33]. The epithelial tumors often invade the basement membrane and often disorganize it [34,35]. 

Dysplasia, or aberrant cell maturation, can be seen in precancerous lesions. Anaplasia occurs while in cancer cells, i.e., when the tumor cells lose their identity as host cells and transform into a more undifferentiated form [33]. Precancerous lesions frequently develop in the duct lumen cavity, and their proliferation is not necessarily accompanied by angiogenesis, which frequently results in apoptosis [36]. Conversely, the overall neovascularization and formation of capillaries in the cancerous tissue are increased, when compared to the precancerous tissue [37]. Apoptosis is another phenomenon that can be used to differentiate cancer cells, as cancer cells often have mutations that disrupt apoptosis leading to cancer initiation [38]. Similarly, when compared to benign tumors, cancer cells have frequently been found to contain a great number of genetic defects and altered gene expression [39]. Altered expression of tumor suppression genes, increased proliferation, increased growth signals, and increased metastasis are examples of specific characteristics of malignant tumors [40,41,42,43].

## 4. Immunotherapy and Cancer

The concept of using the patient’s own immune system as a therapeutic modality in the treatment of neoplastic illness dates back to the eighteenth century. Using immunotherapy to treat cancer made a comeback in the twenty-first century, and substantial progress was achieved in this field. Immunotherapy for cancer has recently brought about revolutionary changes in the area of oncology by extending patients’ chances of survival, even when their disease is at a fatal stage [44]. Owing to the encouraging outcomes, immunotherapeutic drugs have gained greater attention, particularly among clinicians and cancer patients worldwide. Immunotherapy for cancer has the potential to not only cure the primary tumor, but also to prevent metastasis and recurrence. This has led to cancer immunotherapy being a common method of care for those afflicted with the disease [45]. Immunotherapy can work in several ways summarized in Figure 1. Strategies of immunotherapies include use of monoclonal antibodies, checkpoint blockade, vaccines against tumors, adoptive cell therapy, immunotherapy that uses oncolytic viruses and, finally, non-specific immunotherapies.

### 4.1. Monoclonal Antibodies

Monoclonal antibodies (mAbs) are antibodies produced from several copies of a single B cell clone. The term “epitope-specific antibodies” refers to those that recognize and bind to just a small region of an antigen [46]. After the discovery of human–mouse hybrid cell procedures for the production of monoclonal antibodies, these methods were used to generate human-derived hybridomas, which have become a standard in the industrial production of therapeutic antibodies [47]. Unfortunately, the clinical efficacy of the first therapeutic monoclonal antibodies was severely constrained by their rodent origins, which rendered them immunogenic in humans and poor inducers of immunity in patients [47]. The first human research on monoclonal antibody treatment for cancer was a lymphoma patient in 1980 [48]. Researchers in the 1980s began working toward a better understanding of how to create humanized antibodies. Although further study is needed, it is possible to create antibodies that are “completely human” [49].

### 4.2. Checkpoint Blockade

Checkpoint blockade is another class of immunotherapy that utilizes drugs that are checkpoint blockage inhibitors. For instance, humanized monoclonal antibodies that are specific for inhibitory receptors (such as CTLA-4, PD-1, LAG-3, and TIM-3) and ligands (PD-L1) expressed on T lymphocytes, antigen-presenting cells, and tumor cells. By boosting the immune system, they cause an anti-tumor response [50]

### 4.3. Non-Specific Immunotherapies

Non-specific immunotherapies are often known as immunomodulatory treatments. This therapy is not specific to the antigen, and tends to engage both the innate and adaptive immune systems. It involves the following, so as to help the immune system destroy cancer cells: cytokines, like interferons and interleukins, immune-stimulatory substances like CpG oligonucleotides, Bacillus Calmette-Guerin (BCG), antibodies directed against receptors like the agonistic CD40 or inhibitory CTLA-4 antibodies, and enzyme inhibitors, like those directed against cyclo-oxygenase or indolamine-2,3-dioxygenase [51,52].

### 4.4. Immunotherapy Vaccine

Vaccination is the process of artificially introducing substances, known as antigens, into the body to boost the immune system’s defenses against cancer cells. There has been significant research in this field, but more recently, it has been discovered that some vaccinations contain antigens that are very specific to cancer cells and increase survival. These antigens might be pure proteins, DNA, or RNA present in cancer cells, or they can be infectious agents, or pathogens, that have been rendered harmless by heat or chemical treatment [53,54]. Viruses are primarily used as vectors in virus-based cancer vaccines to treat and prevent cancer [55]. Apart from using proteins and genetic material to develop cancer vaccines, the whole cancer cell can also be used [56]. For instance, recently, the GVAX vaccine was developed by genetics to produce the immune stimulatory cytokine granulocyte–macrophage colony-stimulating factor (GM-CSF), a strong immunostimulatory cytokine that increases antigen presentation, activation, and survival of dendritic cells [57].

### 4.5. Oncolytic virus Immunotherapy

Popularly known as virus therapy, this therapy harnesses a genetically modified virus to kill cancer cells. A genetically modified variant of the virus is administered through injection into the tumor. Once it enters the cancer cells, the virus begins its process of self-replication. As a result of this, the cancer cells split apart and die off. When a cell dies, it releases proteins into the surrounding environment. These proteins trigger the immune system to begin targeting any cancer cells in the body that contain the same proteins. Unlike cancer cells, the virus cannot infect cells that are healthy and functioning properly [58]. T-VEC treatment (talimogene laherparepvec), similar to viral vaccination, and commonly referred to as Imlygic, is an example of oncolytic virus therapy. It makes use of one kind of cold sore virus (herpes simplex virus). Some persons with melanoma skin cancer who are unable to have their malignancy surgically removed now have access to T-VEC as a therapy. Trials for head and neck cancer are also examining it. T-VEC is injected directly into the head and neck malignancy or melanoma [59,60].

### 4.6. Adoptive Cell Therapy

Adoptive cell therapy is a form of immunotherapy that entails taking immune-competent cells out of cancer patients and transferring them to other patients. In general, there are three forms of ACT: (1) Lymphocytes that infiltrate the stroma surrounding tumor cells, referred to as tumor-infiltrating lymphocytes, or TILs; (2) Cancer-specific major histocompatibility complex (MHC) molecules that enable T cells to specifically recognize the synthesis, alteration, and processing of certain proteins in cancer cells; (3) The creation of an intracellular, recombinant “immunoreceptor tyrosine activation motif” (ITAM) region and a single-chain variable fragment (scFv) that detects tumor-associated antigen (TAA) recombinants, as the first steps in the synthesis of CAR (Chimeric antigen receptor-) T cells. Then, a recombinant plasmid is transduced into T cells having these two components. Following this stage, the number of T cells increases because this step enables T cells to express the proper tumor surface antigen receptors. Immune cells, known as CAR-Ts, have the ability to recognize and eradicate tumor cells without the need for MHC molecules [61].

## 5. Immunotherapy for Cancer: Overcoming the Challenges

Disease immunotherapies need drugs that work in most people and cancer types. Targetable tumor-specific antigens (TSAs), often called “neoantigens,” generated solely by tumor cells, are a major restriction of cancer immunotherapy [62]. Another option for immunotherapy is focused on tumor-associated antigens (TAAs), which are secreted by both cancerous and normal tissues, but are likely to cause off-target toxicities and have shown very modest success. Many factors have been proposed to explain the wide range of responses seen to cancer immunotherapies among individual patients. These include tumor heterogeneity, differences in kind of cancer and stage, prior treatments, and the immunosuppressive biology of the illness itself [63,64]. In order to choose patients who are likely to have a positive outcome from cancer immunotherapy, it is necessary to first identify biomarkers of predictive or prognostic relevance, which is a difficult and time-consuming process. Only a few prognostic indicators for cancer immunotherapy have been confirmed thus far. Clinical immunotherapy failures may be attributed to tumor heterogeneity and resistant cancer cell clones [65]. There is a high probability of treatment resistance, due to the plasticity and adaptability of cancer signaling networks. As a result of the development of immunotherapy and molecularly targeted drugs, the price of cancer medications and therapies has increased significantly in recent years. To maintain their long-term financial stability, a detailed review of the effects on medical care delivery is required [62].

## 6. Nanotechnology in Cancer Immunotherapy

For cancer immunotherapy to be successful, three things are crucial. To begin with, it is essential for cancer antigens to be successfully transmitted to immune cells, particularly APCs. When cancer antigens and an adjuvant are administered to immune cells, the adjuvant must stimulate an anti-cancer immune response. Thirdly, the immunosuppressive tumor microenvironment (TME) must be regulated for the anti-cancer immunotherapeutic to be effective. Clinical results with cancer vaccinations have been disappointing thus far. To overcome the limitations of conventional cancer immunotherapies, nanoparticles have been intensively explored in the area of drug delivery, due to their capacity to carry medications to target regions effectively, shield pharmaceuticals from proteolytic enzymes, and stay in circulation for prolonged periods of time [45] (Figure 2). Recent developments in nanotechnology have made it possible to load many components, including tiny molecules, peptides, nucleic acids, and cell membranes, onto structures like liposomes, polymer nanoparticles, and inorganic nanoparticles. This makes it possible to co-load antigen and adjuvant in nano-vaccines, ensuring that these active components are administered at the same time to the same APC. Additionally, nano-vaccines promote the effective accumulation of components, such as adjuvant and antigen, in draining lymph nodes and delay their fast spread into circulation [66,67]. Nanoparticle-based vaccinations might, thus, be useful weapons for boosting the immune system and preventing tumor spread [66]. For instance, Song et al. created a nanoplatform for the delivery of an adjuvant and an antigen by covering PLGA nanoparticles with phospholipid membranes. The substantial decrease in the number of metastatic nodules showed that this nano-vaccine might effectively concentrate in lymph nodes and elicit an antigen-specific adaptive T cell response, which reduced the metastasis of B16-OVA melanoma cells [68].

As a consequence of the potential prospects of nanotechnology and immunotherapy for treating cancer metastasis, several inventive and intelligent nanomaterials, including nanorobots, have been developed to enhance therapeutic efficacy. In 2018, Li and colleagues developed a DNA nanorobot that uses DNA origami to deliver payloads precisely to tumors. These nanorobots were able to serve as molecularly sensitive, precise drug delivery systems that delivered thrombin to blood vessels in solid tumors, resulting in intravascular thrombosis and, ultimately, tumor death [69]. DNA origami scaffolds produced by complementary base pairing provided an advanced drug delivery technology that precisely regulated the number and placement of functional moieties, which, in turn, altered drug loading and stimulus-responsive behavior. Only recently, Li and colleagues [70] developed a DNA-based cancer vaccine that was effectively delivered to the lymph nodes that drain tumors and provided tumor antigens to APCs to induce anti-tumor immune responses. The vaccine contained two types of molecular adjuvants, and an antigen peptide, which were put together using a tubular DNA nanostructure. Antigens and adjuvants that were previously imprisoned were made visible as a result of the pH-responsive DNA origami being freed within acidic endosomes. These antigens and adjuvants subsequently attached to their receptors, causing DC activation and antigen presentation, which resulted in T cell activation and cancer cell cytotoxicity. The DNA nanodevice vaccination elicited a strong, tumor-specific T cell immune response that, subsequently, caused the tumors in mice to shrink, as well as an extended T cell immunological memory response that markedly protected the animals from tumor metastasis [70,71].

The use of nanomedicines has exciting prospects for boosting the efficiency of such vaccinations. Different nanoplatforms, like Immunoliposome, gold nanoparticle, iron oxide, and PLGA nanocarrier, have been studied for their potential to deliver and enhance anti-tumor immunity and decrease unwanted side effects by transporting molecular, cellular, or subcellular vaccines to lymphoid tissues and cells [45,72]. Table 2 enumerates examples of various such nanoparticles explored in the immunotherapy of cancers. The different applications of nanotechnology in immunotherapy include nanoparticles used to deliver tumor antigen, adjuvants and TME immunomodulators.

### 6.1. Nanoparticle-Based Delivery of Anticancer Antigen

It is necessary for tumor antigens to be efficiently delivered to APCs in order to develop tumor immunity. The use of nanoparticles as delivery systems for securely delivering tumor antigens to lymph nodes has been investigated [94]. The transport of nanoparticles to lymph node targets is highly dependent on particle size, surface charge, shape, and hydrophobicity. The Extracellular matrix (ECM) can capture large nanoparticles, while medium-sized nanoparticles may stay in the bloodstream and successfully reach the lymph nodes through lymphatic capillaries. In contrast to large-size nanoparticles, small nanoparticles may leak out of blood arteries during circulation. However, the medium-sized (5–100 nm) nanoparticle size is ideal for effectively delivering tumor antigens to the lymph nodes. In order to distribute nanoparticles more precisely, it is also feasible to use active transport by adding chemical ligands to nanoparticles, such as mannose. Cellular internalization and the initiation of the immune response are significantly influenced by the carrier’s surface charge [95] (Figure 3).

### 6.2. Nanoparticle-Mediated Adjuvant Delivery

Adjuvants are molecules that boost the immunogenicity of tumor antigens, which might be deficient when delivered on their own. In order to boost the body’s natural ability to fight cancer, adjuvants are often employed in cancer immunotherapy [96]. A wide variety of blood and solid cancers like leukemia, melanoma and lung cancers are treated effectively using nanoparticles. Tumor development was inhibited, and survival was markedly increased when tumosomes were injected into mouse tumor models. Tumosomes include two immunostimulatory adjuvants: DDA, which acts as a cell-invasion moiety, and two malignant membrane proteins (cancer antigens), MPLA, which serves as a warning signal. The therapeutic effectiveness of this approach might be further improved by combining it with other treatment methods, such as cell-based therapy, gene therapy, and chemotherapy [83,91,97].

### 6.3. Nanoparticle-Mediated Modulation of the Immunosuppressive TME

Through the production of an immunosuppressive TME, tumors may encourage the proliferation and dissemination of cancer cells. The immunotherapy treatment for cancer often includes the alteration of this environment as a significant approach [98]. Tumor-associated macrophages are abundant in the TME and impede anti-cancer immune responses by generating inflammatory cytokines, such as IL-12, IL-1b, TNF-α, and IL-6. Furthermore, effectively suppressing, or even eradicating, regulatory T cells, may induce anti-tumor immunity. Immune cell activation, maturation, and differentiation may all be inhibited by TGF-β, a cytokine that is overexpressed in breast, liver, and lung cancer [45]. Myeloid-derived suppressor cells (MDSCs) are a kind of tumor-suppressor cells that are often discovered in the tumor microenvironment (TME) of several malignancies, including those of the breast, lung, gastrointestinal tract, and liver. MDSCs secrete indoleamine 2,3-dioxygenase (IDO), IL-10, ARG1, and NOS2 activate regulatory T cells (Tregs) and repress other immune cells. In mice with metastatic melanoma, TGF-α inhibitors, that were encased in lipid nanoparticles, were recently administered to activate both innate and adaptive immune responses, which led to the inhibition of tumor development and an improvement in survival rates. This method is an innovative approach to addressing the limits of the currently available cancer immunotherapy.

## 7. Cancer Immunoprevention and Its Strategies

Cancer prevention and cancer treatment are the two main interventions to target cancer. Cancer immunoprevention is different from cancer immunotherapy, as immunotherapy targets the immune system to treat an existing disease, whereas immunoprevention modifies the immune system for disease prevention. As discussed earlier, there are several drawbacks of using immunotherapy in cancer treatment, like the development of tolerance and escaping detection by the immune system [99]. Cancer immunoprevention is based on the principle that the immune system controls the onset and progression of cancer, hence, modulation of the immune system to provide enhanced immune response could reduce the cancer risk in healthy individuals. Prevention of cancer aims to reduce incidence of cancer and there are primarily two approaches: chemoprevention and immunoprevention. Chemoprevention utilizes drugs and natural compounds that prevent cancer [100]. Drugs like tamoxifen and raloxifene have been approved by the FDA for the prevention of breast cancer [101,102]. Celecoxib has been approved for colorectal cancer (CRC) prevention in FAP [103,104] and valrubicin has been approved for bladder cancer [105]. Conversely, immunoprevention directly or indirectly targets the immune system to prevent cancer. The immune system is well known to be crucial in the defense against infectious diseases. It does, however, play a crucial part in cancer prevention. The involvement of the immune system in tumor prevention has long been speculated, but it was only recently proven, when it was shown that mice with compromised immune systems eventually grew tumors, while immunocompetent mice of the same age did not [106,107].

Immunoediting is a process where the immune system can either suppress or promote cancer. Tumors often cause this immunoediting to enable its progression [108]. Loss of MHC Class I expression, T cell anergy, and expression of the inhibitory receptors PD-1, LAG-3, and TIM-3 are all caused by the immune-suppressive mechanisms in the tumor microenvironment, which contribute to T cell exhaustion [109]. Hence, strengthening immune response before the cancer progression, using various immunoprevention strategies, could be a solution to this. Numerous strategies of immunoprevention have been employed in cancer prevention in recent years. The strategies range from strategies that target infectious agents that cause cancer, like HPV, and immunomodulators to nanoparticle-based drug delivery and nanoparticle-based targeting of tumor antigens.

### 7.1. Vaccines in Cancer Immunoprevention

Vaccines have been established as one of the most prevalent therapies to prevent infectious diseases over the years. Vaccines induce endogenous effective and memory immune responses by enhanced antigen presentation. The ease of the delivery system, limited side effects and, most importantly, induction of long-term immune responses are some critical advantages of vaccines in disease prevention. Similar to targeting infectious diseases, vaccines have been used in immunotherapy against cancer, and recent literature has provided evidence of application of vaccines in immunoprevention of cancer like melanoma and colon cancer [110,111,112]. The vaccines used in cancer immunoprevention could be peptide-based, cell-based or genetic vaccines, each having its own merits and demerits. Peptide vaccines, for example, are economical and easy to produce [111,113,114]. The fact that cell-based vaccinations are generated from tumor cell lines and, thus, represent both known and unidentified tumor antigens is an advantage [113]. Immune cell vaccines that use dendritic cells loaded with specific tumor antigen, mRNA derived from tumors and tumor cell lysate have also been used [114,115]. Moreover, genetic vaccines that use DNA or RNA to express tumor antigens delivered by viral vectors have also been developed [116,117]. A Mucin (MUC1)-based vaccine was employed in a preclinical model of colon cancer by Mukherjee et al. MC38 colon cancer cells, expressing MUC1, were put into immune-competent MUC1-tolerant hosts, who were then given the vaccination. The loss of MUC1 tolerance resulted in a potent anti-tumor response [110]. Similar results were observed by Kimura et al. In patients without cancer but having premalignant lesions like colonic adenomas, that grow to become melanoma, high immune response on administration of a vaccine based on the tumor associated antigen MUC1 was demonstrated. Such vaccines are safe, immunogenic and elicit long term memory, which is crucial for cancer prevention [111]. In a different placebo-controlled phase II experiment, women with low grade uterine cervix premalignancy were given the HPV vaccine. Due to the immunization, a significant immunological response and activation of long-term memory T cells were seen [112].

### 7.2. Immunoprevention and Virally-Induced Tumors

As is well known, a variety of viruses have been linked to the development of cancer. It is known that the hepatitis C virus causes hepatocellular carcinoma (HCC) [118], the Epstein-Barr virus causes Burkitt’s lymphoma and nasopharyngeal carcinoma [119], the human T-cell lymphotropic virus causes adult T-cell leukemia [120], human herpesvirus 8 causing Kaposi’s sarcoma [121] and *H. pylori* causes gastric cancer [122]. Many of these cancer-causing organisms do not have vaccines, and, thus, there is an urgent need for treatment against them. One-sixth of all human malignancies are caused by infections, making them excellent candidates for cancer prevention [123]. Various cancer immunoprevention vaccines have also been approved by the FDA for virally induced cancers. Vaccines against human papillomavirus (HPV), which is responsible for 70% of cervical cancers, are an example of these, and another example is the vaccine against hepatitis B virus (HBV), which causes HCC [124,125,126]. 

The two vaccinations authorized to prevent HPV infection are Gardasil and Cervarix. The L1 protein of HPV16 and 18 have virus-like particles present in both of these vaccines. This protein is crucial, as it is involved in viral entrance [127,128]. A recent study observed an 83% reduction in HPV16 and 18 infection prevalence and a 51% decline in cervical intraepithelial neoplasia (premalignant lesions) after a few years of vaccination [129]. Another study conducted by Lei et al. revealed that giving the HPV vaccine to girls between the ages of 10 and 30 reduced the likelihood of developing premalignant lesions and was also linked to a lower risk of developing invasive cervical cancer [130]. Despite these encouraging findings, there have been some setbacks in the immunoprevention of virally generated cancer. One instance of this was the existence of seven HCV genotypes with a wide genetic variety and a high probability of viral replication error, which restrict the creation of an effective vaccine [131]. 

### 7.3. Tumor Antigens in Cancer Immunoprevention

TAA and TSA are the two types of tumor antigens. TAA are antigens that are overexpressed in cancerous cells but expressed normally in healthy ones. On the other hand, TSA are antigens that are only expressed in cancerous cells. TSA are expressed as a result of somatic mutations or viral oncogenes [132,133,134]. By generating a particular T-cell response against cancer cells, these antigens can be employed as a target for immunopreventive treatments for non-virally developed cancers [135]. There are two basic methods used to identify these cancer antigens. Direct immunology is accomplished by isolating tumor-directed T cells and locating the cDNA encoding the Tc cell epitopes from cDNA expression libraries [136]. The second technique, indirect immunology, uses an algorithm-based strategy to predict tumor antigen binding to MHC molecules [137]. However, validation is required because it is possible that the anticipated epitopes will not be naturally digested and presented by MHC molecules [138]. The creation of immunopreventive anti-tumor vaccines can be done using either the direct or indirect immunology method. The optimal tumor antigen for use in cancer immunoprevention might contain immunogenetic properties, as well as an oncogenic function. Additionally, external, rather than internal, expression of the antigen on tumor cell surfaces is preferred [139,140,141]. Class I oncoantigens, for instance, are extracellular tumor proteins that are targets for both humoral and cellular defense and are, thus, a prime candidate for immunoprevention strategies. However, some issues with these antigens include immunoediting and the emergence of antigen loss variants [142,143,144]. The antigens expressed on premalignant lesions can also be targeted for immunoprevention in a manner similar to targeting tumor antigens. Premalignant lesions express several tumor antigens, including MUC1, Carcinoembryonic Antigen (CEA), and Human Epidermal Growth Factor Receptor 2 (HER2) [144,145,146].

MUC1 is expressed in normal epithelium tissues. However, an abnormally glycosylated form is observed in tumors, like those of pancreatic, breast, and colon cancer [147,148,149,150,151,152]. Premalignant tumors, like colon polyps and pancreatic neoplasms, that progress to form CRC and pancreatic ductal adenocarcinoma have been reported to overexpress MUC1 antigen and, hence, it is a potential antigen to target for immunoprevention [153,154]. Breast, lung, stomach, and CRC are among the tumors when carcinoembryonic antigen is overexpressed [155,156]. Furthermore, it has been overexpressed in precancerous lesions of CRC [157,158]. HER2, an antigen that is expressed in large quantities in breast cancer, and is similar to MUC1 and CEA, was also found to be overexpressed in premalignancy stages of breast cancer [159,160].

### 7.4. Immunomodulators in Cancer Immunoprevention

Some drugs are capable of inducing, or repressing, the immune system in a nonspecific way against a specific target. BCG, IL-2 and imiquimod are examples of such FDA approved immune-modulators for cancer immunotherapy [161,162,163]. Apart from cancer immunotherapy many immunomodulators, either alone or in combination with other interventions, are now finding their application in cancer immunoprevention. The immunomodulator, imiquimod, which is a TLR7 agonist, was observed to be effective in the treatment of premalignant lesions of skin squamous cell carcinoma (SCC) [164]. As was previously mentioned, vaccinations targeting MUC1, which is elevated in polyps, and denotes an elevated risk of CRC, have been developed. As an adjuvant, a TLR3 agonist was also added to the vaccine; this addition had no apparent high-grade side effects, and 43% of the patients later acquired IgG antibodies against MUC1 [111]. Another illustration of this is the multiple myeloma vaccination, PVX-410, which is given both alone and in conjunction with lenalidomide [165]. Lenalidomide, a thalidomide derivative, licensed by the FDA to treat multiple myeloma, has anti-tumor activity in addition to being an immunomodulator that increased T cells’ and NK cells’ activities [20,166,167]. The percentage of CD3+CD8+ T cells that were tetramer-positive and IFN-γ-positive increased more than a fold, on average, as a result of the combination, according to Nooka et al. [165]. Imiquimod, a TLR7 agonist, is an FDA-approved immunomodulator that induces innate and adaptive immune responses by secreting cytokines in SCC, a premalignant lesion known as actinic keratosis [[168,169]. Advanced clinical trials using imiquimod 2.5% and 3.75% creams for three weeks produced complete AK clearance rates of 43.2% and 47.9% at the one-year follow-up [170].

### 7.5. Immune Checkpoint Inhibitors in Immunoprevention

In recent years many treatments that involve immune checkpoint inhibitors have been approved and this strategy has now been applied to the field of cancer immunoprevention. As discussed earlier, immune checkpoints, like PD-L1 and CTLA-4, are present on normal cells. PDL1 and CTLA-4 bind to their respective receptors, PD-1 and B7, on T cells and this inactivates the immune response against them. Cancer cells use this property to evade immune response. Immune checkpoint inhibitors block this response, and this activates the immune system. Several precancerous lesions, in addition to malignancy, displayed the expression of PD-L1 associated with transformation into cancer [171,172]. For example, oral precancerous lesions showed PD-L1 expression [171,172,173]. In a mouse model, progression of an oral premalignant lesion induced by a carcinogen was stopped by blocking the PD-1/PD-L1 pathway [173,174]. A preliminary investigation into the prevention of squamous cell carcinoma by PD-1 inhibition in a murine model yielded encouraging findings [174]. A total of 33 individuals with oral proliferative verrucous leukoplakia are currently being tested in a clinical trial using nivolumab, an anti -PD-1 antibody that blocks the PD-1/PD-L1 pathway (NCT03692325). PD-1 inhibition may be effective in preventing oral SCC, although patients should be closely watched for the emergence of irAEs.

### 7.6. Nanoparticle-Based Cancer Immunoprevention

Nanotechnology is an extremely wide and versatile field that has been useful in many disciples in an innovative and unpredictable way. The previous section of this review explained how nanoparticles have been used in various facets of cancer immunotherapy and have given very positive outcomes (Figure 4). Although there is little current research on cancer preventive treatments using nanotechnology, they are indeed possible. Immunoprevention has developed a lot in recent years. However, some drawbacks do exist. One illustration of this is the minimal cross-presentation and substantial degradation via the endocytosis pathway that tumor specific neoantigen based cancer vaccines experience, despite their efficacy in cancer immunoprevention. A thiolated nano-vaccine that enabled the direct cytosolic administration of neoantigen and Toll-like receptor 9 agonist CpG-ODN was recently developed using nanotechnology. This nanovaccine could bypass lysosome breakdown and improve neoantigen absorption and local concentration, which activated antigen-presenting cells and boosted anti-cancer T-cell immunity [175]. Another study found that the delivery of nutraceuticals, like curcumin, using nanoparticles reduced the chronic inflammation that is known to cause cancer [176]. The following section of this review provides details of several examples of applications of nanotechnology in cancer immunoprevention.

## 8. Nanotechnology and Cancer Immunoprevention

Advances in nanotechnology are now finding applications in numerous facets of immunoprevention. Several examples of the same are illustrated in this section, enumerated in Table 3. Nanotechnology was used to create nanoparticles derived from ginger that had ginger bioactive compounds. In colitis models, oral uptake of these particles decreased the pro-inflammatory cytokines (TNF-α, IL-6, and IL-1), raised the anti-inflammatory cytokines (IL-10 and IL-22), and was shown to be a preventive therapy against colitis-related malignancy [177]. In colitis models, oral GDNPs 2 administration increased IEC survival and proliferation, decreased pro-inflammatory cytokines (TNF-α, IL-6, and IL-1), and increased anti-inflammatory cytokines (IL-10 and IL-22). These results indicated that GDNPs 2 might have the ability to attenuate harmful factors, while promoting the healing effect. In conclusion, GDNPs 2, nanoparticles made from edible ginger, represent a novel, all-natural method for enhancing IBD prevention and therapy, while also eliminating drawbacks, like potential toxicity and the small manufacturing scale that are typical of synthetic nanoparticles [177]. Iron NPs and iron oxide NPs have also been used to encapsulate tumor antigens like ovalbumin and CEA to induce anti-tumor responses [86]. Other nanoparticles that are extensively used in cancer immunoprevention are liposomes, that are loaded with numerous antigens, like CpG-ODN, TGF-β inhibitors [178,179,180,181,182,183].

Another study administered selenium nanoparticles, containing Lactobacillus plantarum, in a breast cancer murine model two weeks prior to induction of tumor. It was observed that the number of proinflammatory cytokines and NK cell activityincreased, followed by increased survival in the test mice [192]. As discussed in previous sections, tumor antigens are extensively used in cancer immunoprevention. Recently, a group of researchers used nanoparticles to deliver the tumor antigen particles to induce T cell responses. A biodegradable nanoparticle PLGA-NP, containing the peptides hgp_10025-33_ and TRP2_180-188_, associated with murine melanoma, was developed. It was found that immunization with these nanoparticles injected subcutaneously in animal models considerably slowed the development of B16 melanoma cells [193]. Nanotechnology has also been used for immunomodulation for prevention of tumors like melanoma, colon carcinoma, lymphoma etc. [86,180,188]. A study used nanoscale liposomal polymeric gels to deliver TGF-β inhibitor and IL-2 to the tumor environment. This delayed tumor growth and increased survival [181]. Another study coated graphene oxide (GO) with a photosensitizer to operate as a tumor integrin v6-targeting peptide (the HK peptide). This nanoparticle boosted the immune system’s ability to fight cancer and stopped the tumor from returning [194]. Avasimibe, with a multi-peptide Kras vaccination, improved T cell infiltration at the tumor site and slowed the spread of lung cancer. This course of treatment is a brand-new lung cancer immunoprevention strategy [195].

## 9. Applications of Nanotechnology in Cancer Vaccines

It is established that nanotechnology has been applied to various strategies of immunotherapy and immunoprevention. It has several advantages, such as improving access to the lymph nodes, better tracking, and improved antigen presentation, further leading to increased anti-tumor immune response. Immunoprevention seeks to halt the progression of cancer, and research is being conducted to assess the viability of utilizing the theoretical underpinnings of immunoprevention for cancer types that are not linked with infectious agents [109]. Although immunomodulation, and antibodies are also emerging cancer prevention strategies that are being investigated, prophylactic cancer vaccines are the most effective cancer preventive strategy [109]

The cancer vaccines work in delivery of distinct components, like antigens, adjuvants or antigen presenting cells. They could either act like an adjuvant themselves or elicit an immune enhancing ability. Various components, like mRNA, subunits, peptides, DNA, neoantigen and even whole cells, are used as antigens in the preparation of cancer vaccines [196]. Table 4 enumerates a number of different nanoparticles that have successfully induced and increased anti-cancer immune responses in various cancer types, like melanoma, lymphoma, breast cancer, colon cancer etc. Furthermore, many properties of nanoparticles are also crucial in considering best outcomes. Properties like particle size, rigidity, surface charge, targeting ligand, and, finally, immunomodulatory agent added all contribute to the efficiency of the nanoparticle used in the development of cancer vaccines [196]. 

Numerous types of nanoparticles are being used in development of distinct cancer vaccines that target some specific part of the immune system response and elicit an anti-cancer response. Liposomes are one such category of nanoparticles, that have been demonstrated to easily pass through the lipid membrane of various immune cells and cause their activation [199,200,201,203]. Nanoparticles made of inorganic materials like gold and aluminum are another class of nanoparticles that are quite popular, due to their nontoxic and immunologically inert nature. [204,205,208,212]. Polymeric NPs made up of Chitosan, PGLA etc. are also frequently used in cancer vaccine production [193,209]. These nanoparticles can contain various active compounds that could impart prophylactic or therapeutic effect.

### 9.1. Nanotechnology in Peptide-Based Vaccines

Tumor eradication necessitates the production of MHC I-restricted cytotoxic T lymphocytes (CTLs). This is accomplished by delivering TAA as a peptide or gene in conjunction with strong activation of DCs, which can then stimulate TAA-specific T cells. Trp2, has been identified as a melanoma TAA and has been tailored to various nano-platforms [80,213]. Xu et al. [80] devised a polyplex preparation by varying the ratios of arginine-modified Trp2 and CpG. Furthermore, co-encapsulation of Trp2 peptide and CpG co-within lipid calcium phosphate nanoparticles (LCP NPs) leads to efficient delivery into DCs, thereby reducing the tumor burden [80]. DCs are known to better phagocytose cationic nanoparticles (CNPs) when compared to other cell types. The antigen-presenting ability, together with the immunostimulatory properties, of DCs efficiently initiates T cell responses, and triggers rapid uptakes of CNPs, thereby boosting the immunogenicity of cancer vaccines [214,215]. PLGA polymeric nanoparticles are indeed encouraging TAA delivery platforms. This is essential when addressing TLR7/8 agonist delivery, including peptide/protein based TAAs, which are generally confined by limited retention at the administered region. PLGA nanoparticles are a desirable delivery platform for such TLR agonists as they proficiently enter endosomes/lysosomes upon cytosolic delivery [216].

### 9.2. Nanotechnology in Nucleic Acid-Based Vaccines

Only a small percentage of patients experienced mild therapeutic effects when using peptide-based antigens as cancer vaccines [217]. The combination of genomic sequencing and nanotechnology has enabled the creation of effective, reliable, and personalized DNA or mRNA vaccines against specific TAAs [218]. Nonviral pDNA or mRNA vaccines delivered via nanocarriers are safer and more cost effective than traditional vaccines. This idea was successfully proved with lipidoid nanoparticles [217]. The fact that pDNA/mRNA vaccines elicit both CTLs and helper T cells simultaneously, via both MHC class I and II pathways, is a significant benefit [211,219]. Numerous polymer and lipid platforms were employed to complex with pDNA for therapeutic vaccine applications to enable expression. Chitosan [220], PLL [218], and PEI [221] are examples.

### 9.3. Nanotechnology in Tumor Cell or Lysate-Based Vaccines

Applications of tumor cell components, like membranes, in the development of nanoparticle-based cancer vaccines is gaining increased attention, due to their ability to mimic the characteristics of the tumor cells. Polymeric nanoparticles, when coated with a layer of membrane coating derived from tumor cells, presented a plethora of tumor antigens and promoted tumor specific immune response [187]. Another study utilized artificial antigen presenting cells that are coated with human leukocyte antigen–immunoglobulin fusion protein (HLA-Ig) and CD28-specific antibody. These particles were able to activate tumor specific immune response in melanoma cell lines. Further, T cell responses were also enhanced [188].

## 10. Future Prospects and Challenges in Cancer Immunoprevention

We understand that cancer immunoprevention has succeeded in treating virally induced tumors due to the ease of detecting premalignant lesions in this tumor. Furthermore, it is possible to target the HPV oncogenic peptides without seriously harming normal cells. However, non-virally caused cancers are more difficult to cure since they are hard to find. Similarly, detection of tumors like breast cancer is easy as the stepwise progression of these tumors is understood, such as identification of premalignant lesions and the mutations involved [222]. Another challenge of cancer immunoprevention is the criteria for the decision of the target individual at high risk to tumor progression. Moreover, the state of the tumor in the target individual needs to be of a premalignant type [222,223]. Another concern with these methods is their possible adverse effects, and since the target person is healthy, a risk–benefit analysis should be taken into account while developing preventive measures [16]. However, immunoprevention is far safer with fewer side effects and also provides long-term protection due to immunological memory, in comparison to surgical removal or chemoprevention techniques, for the prevention of cancer. Cancer immunoprevention has been shown to be effective in preclinical and early clinical investigations; however, this approach is still in its early stages and has to be accessed in more advanced randomized clinical trials before becoming a standard of care.

## 11. Conclusions

The immune system has a critical role in the progression of tumors and its different components can both promote and suppress tumor growth. Cancer immunotherapy and immunoprevention are strategies to treat and prevent cancer. This review has discussed various strategies of cancer immunotherapy and immunoprevention and has, further, highlighted the advances of nanotechnology in enhancing the efficiency of these strategies. Finally, as cancer vaccines are one of the most popular methods of immunotherapy and immunoprevention, we need to understand the current status of cancer nanotechnology-based cancer vaccines and their safety. The study of these new technological advances in early diagnosis could be of benefit in the identification of high-risk individuals for successful prevention of tumors at an early stage.

## Figures and Tables

**Figure 1 vaccines-10-01727-f001:**
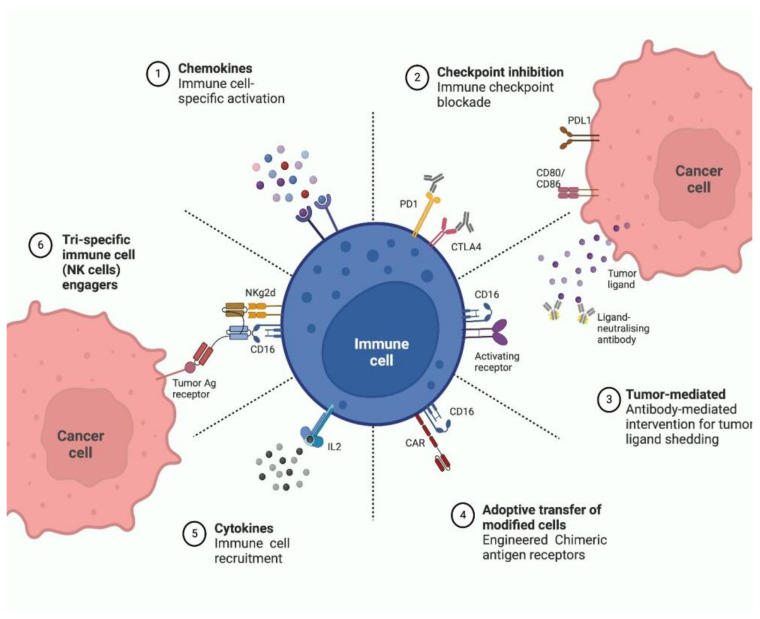
Approaches for immune cell-based cancer therapy.

**Figure 2 vaccines-10-01727-f002:**
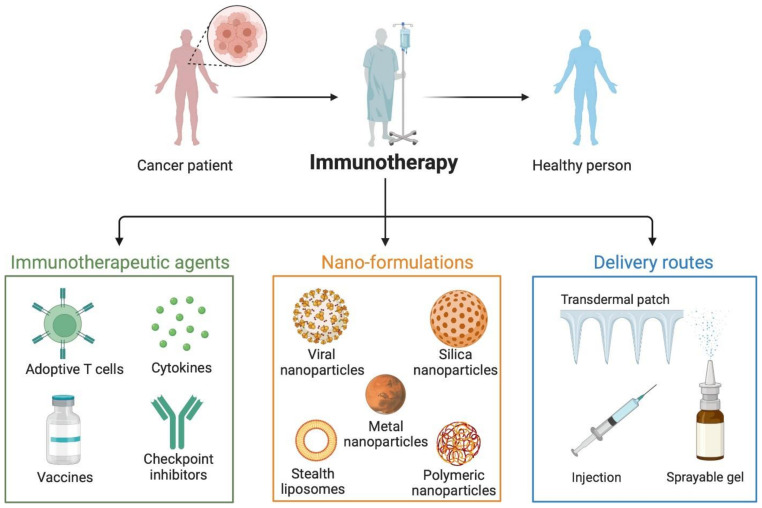
Various nanomedicine enhanced cancer immunotherapy strategies.

**Figure 3 vaccines-10-01727-f003:**
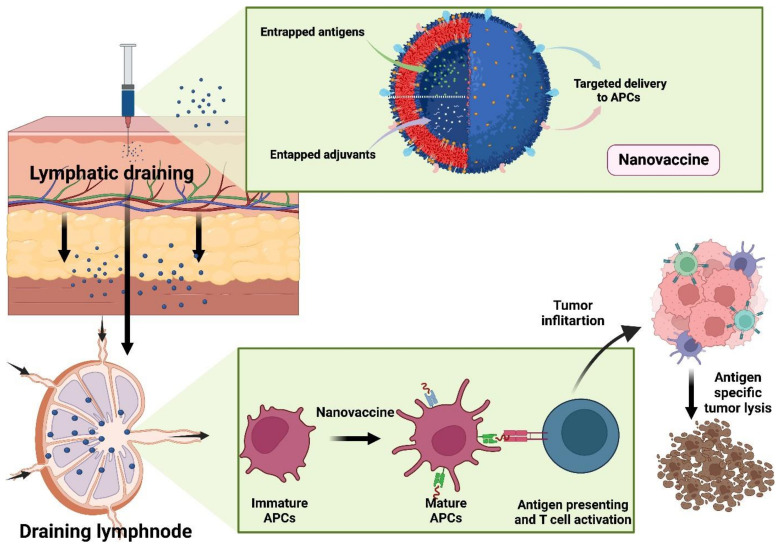
Nano vaccine and cancer immunotherapy.

**Figure 4 vaccines-10-01727-f004:**
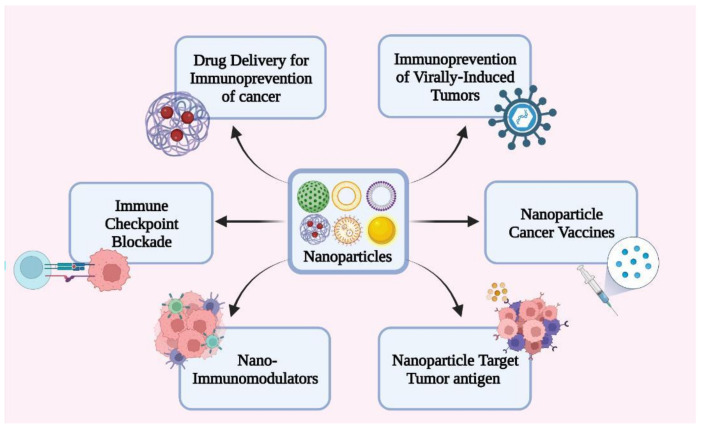
Applications of Nanotechnology in Cancer Immunoprevention.

**Table 1 vaccines-10-01727-t001:** Distinguishing Premalignant from Malignant Lesions.

Characteristics	Premalignant Lesions	Malignant Tumors
Genetic abnormalities	Few	Many
Apoptosis	Partially effective	Ineffective
Tumor suppressive genes	Partially active	Inactive
Cell proliferation	Normal	Increased
Stem and progenitor cells	Semi-increased	Increased population
Invasion status	Noninvasive lesion	Invasive
Basement membrane	Intact	Breached and disorganized
Cell morphology	Dysplasia: similar to the tissue of tumor origin	Anaplasia: revert to undifferentiated form
Neovascularization	Normal	Increased

**Table 2 vaccines-10-01727-t002:** Advances of Nanotechnology in Immunotherapy.

Nanoparticle	Active Agent	Delivery Method	Cancer Type	Effect/Inference	Clinical Trial Status	References
CNT-CpG	CpG ODN	i.tm.	Subcutaneous Melanomas	Eradicated glioma and increased tumor immunity	Pre-clinical (in vivo study)	[73]
CNT	Tumor lysate		Human NSCLC	Promoted lymphocyte mediated cytotoxicity by NF-ΚB	Pre-clinical (in vitro study)	[74]
HS-TEX	Chemokines (CCL2, CCL3, CCL4, CCL5, and CCL20)	i.tm.	Lung and skin cancer	Increased activation of T cell and dendritic cells	Pre-clinical (in vivo study)	[75]
AuNPs	CpG ODN	i.tm.	B16 melanoma	Promoted macrophage and dendritic cell invasion into tumor, inhibited tumor growth and increased survival.	Pre-clinical (in vivo studies)	[76,77]
Hyaluronic acid	CpG ODN	i.tm.	Lymphoma	Enhanced anti-tumor activity and immune memory	Pre-clinical (in vivo study)	[78]
Iron Oxide NPs	CpG ODN	i.p.	Colon cancer	Increased t cell responses and decreased tumor growth	Pre-clinical (in vivo study)	[79]
Liposomes	Trp2 peptide	i.v.	B16 melanoma and lung metastasis	Enhance T cell responses	Pre-clinical (in vivo study)	[80]
Polymeric NPs (PC7A NP)	Ovalbumin	i.v.	Melanoma, lung, and colon tumor	Improved delivery of tumor antigen, increased surface presentation and inhibited tumor growth	Pre-clinical (in vivo study)	[81]
Oligonucleotide Nanoring	Anti-Bmi1 and anti-Mel 18 shRNA with CpG ODN	i.tm.	Medulloblastoma	Inhibited tumor proliferation and growth	Pre-clinical (in vivo study)	[82]
Liposomes	E7 peptide	s.c.	Lung cancer	Activate antigen presenting cells and stimulate DCs	Pre-clinical (in vivo study)	[83]
R8-Lip	α-galactosylceramide	i.v.	Lung cancer and malignant B16 melanoma	Activated NK cells and increased anti-tumor immune reesponse	Pre-clinical (in vivo study)	[84]
PLGA-NPs	TRP2180-188 and 7-acyl lipid A	s.c.	B16 Melanoma	Induced interferon secretion, activated T cell responses, and decreased tumor size.	Pre-clinical (in vivo study)	[85]
Polymeric NPs	CpG ODN	i.d.	B16 Melanoma	Activated DCs and inhibited tumor growth	Pre-clinical (in vivo study)	[86]
Protein cage NPs	Ovalbumin	i.v.	B16 Melanoma	Activated cytotoxic T cells and suppressed tumor growth	Pre-clinical (in vivo study)	[87]
Cowpea mosaic virus nanoparticles		i.t.	Melanoma, colon, breast, lung and ovarian cancer	Prevented lung melanoma and generated anti-tumor immunity	Pre-clinical (in vivo study)	[88]
CHP nanogel	Truncated 146HER2 protein	s.c.	HER2 expressing tumor patients	Induced HER2-specific humoral responses in patients with HER2-expressing tumors	Phase I	[89]
Liposomes	RNA encoding tumor antigens	i.v.	Melanoma	Induced effector and memory T cell responses, caused INF-α release from macrophages,	Phase I	[90]
Virus-like NPs (MelQbG10)	Melan-A/MART-1 Peptides with Montanide and Imiquimod	i.ln	Melanoma (Stage III-IV)	Enhanced memory and effector CD8+ T-cell responses	Phase IIa	[91]
Virus-like NPs	Melan-A/MART-1 Peptides	i.d	Melanoma (Stage II-IV)	Increased antigen presentation to DC cells and enhanced T cell responses	Phase IIa	[92]
Exosomes	MAGE 3 peptides	i.d.	Melanoma (Stage III-IV)	Promoted tumor rejection and increased T cell responses	Phase II	[93]

Abbreviations: i.v.: intravenous injection; i.d.: Intradermal injection; i.p.: intraperitoneal injection; i.g.: intragastric; i.t.: intratracheally; s.c.: subcutaneous; i.ln: Intra-lymph node injection; i.tm.: Intratumor, CHP: cholesteryl pullulan; NSCLC: Non-Small Cell Lung Carcinoma; MAGE: melanoma associated antigen; HS-TEX: exosomes derived from heat-stressed tumor cells; R8-Lip: Stearylated octaarginine-modified liposomes; CpG ODN: CpG oligodeoxynucleotides; PGLA: poly(lactic-co-glycolic acid); TrP2: Tyrosinase-related protein 2; Phase IIa trials: Trials that include administrating different quantities of drug to check for dose response relationship; Phase II (Phase IIb): Trials that determine the efficacy of drug in preventing, diagnosing and treating a disease.

**Table 3 vaccines-10-01727-t003:** Applications of Nanotechnology in Cancer Immunoprevention.

Nanoparticle	Active Agent	Delivery Method	Cancer Type	Effect/Inference	Clinical Trial Status	References
Iron oxide beads	Ovalbumin	s.c.	B16 Melanoma	Induced CD8 dependent protective immunity in vivo	Pre-clinical (in vivo study)	[184]
Polystyrene microspheres	Ovalbumin	s.c.	T cell Lymphoma	Protected against tumor growth and treated existing tumors	Pre-clinical (in vivo study)	[185]
LPH-NPs	TGF-β si-RNA	i.v.	Melanoma	Knockdown of TGF-β and inhibited tumor growth by 52%. Increased activity of cytotoxic T cell and decreased level of T regs cells	Pre-clinical (in vivo study)	[178]
Iron oxide-zinc oxide NPs	CEA	i.v.	colon adenocarcinoma	Enhanced T cell responses, reduced tumor growth and better survival	Pre-clinical (in vivo study)	[79]
γ-PGA NPs	Ovalbumin	Nasal		Induced antigen specific cellular and humoral immunity	Pre-clinical (in vivo study)	[186]
Liposomes	CpG-ODN	i.m.	B-cell lymphoma	Induced strong cellular and humoral immunity	Pre-clinical (in vivo study)	[179]
Cationic liposomes	CpG	i.d.	Melanoma	Increased DC maturation	Pre-clinical (in vivo study)	[180]
Liposomal polymeric gels	Cyclodextrins, TGF-β inhibitor and IL-2	i.tm.	Melanoma	Delayed tumor growth and increased tumor survival	Pre-clinical (in vivo study)	[181]
Cationic liposomes	TLR agonist (CpG ODN) and Ovalbumin	s.c. or i.d.	Melanoma	Increased antigen presentation and enhanced T cell responses	Pre-clinical (in vivo study)	[182]
Cationic liposomes	α-GalCer with CpG and Ovalbumin	s.c.	B16 Melanoma	Increased activation of NK, DC and T cells	Pre-clinical (in vivo study)	[183]
Tumor cell membrane coated PLGA NPs	Ovalbumin and PAM or CpG		Melanoma	Increased antigen presentation and immune responses	Pre-clinical (in vitro study)	[187]
Tumor cell membrane coated NPs	HLA-Ig and anti-CD28		Melanoma	Promoted tumor specific immune response and induced antigen specific activation of T cell	Pre-clinical (in vitro study)	[188]
Latex beads	Trp2 peptide and CpG	s.c. and i.v.	Melanoma	Inhibited tumor growth and enhanced T cell responses	Pre-clinical (in vivo study)	[189]
iron-dextran particles and quantum dot nanocrystals	HLA-Ig and anti-CD28	i.p and i.v	Melanoma	Generation of antigen specific cytotoxic T lymphocytes	Pre-clinical (in vivo study)	[190]
aAPCs	Trp-2 peptide	i.v.	Melanoma and lung metastasis	Enhanced T cell responses and reduced tumor growth	Pre-clinical (in vivo study)	[191]

Abbreviations: LCP-NPs: lipid-calcium-phosphate (LCP) nanoparticle; LPH-NPs: liposome-protamine-hyaluronic acid nanoparticle; γ-PGA: γ-polyglutamic acid; α-GalCer: α-galactosylceramide; aAPC: artificial antigen presenting cells.

**Table 4 vaccines-10-01727-t004:** Advanced of Nanotechnology in Cancer Vaccines for Immunotherapy and Immunoprevention.

Nanoparticle	Active Agent	Delivery Method	Cancer Type	Effect/Inference	Clinical Trial Status	References
**Au-NPs**	Mangiferin	i.v.	Prostate cancer	Enhanced levels of anti-tumor cytokines with reduced pro-tumor cytokines	Pre-clinical (in vivo study)	[197]
**GDNPs 2**	Ginger bioactive constituents	Oral and i.p.	Colitis-Associated Cancer	Control immune response and chronic inflammation	Pre-clinical (in vivo study)	[177]
**Se-NPs-enriched Probiotic**	Lactobacillus plantarum strain	Oral and i.v.	Breast cancer murine	Levels of proinflammatory cytokines increased and increased NK cell activity. Decreased tumor volume and increased survival	Pre-clinical (in vivo study)	[192]
**Thiolated nano-vaccine**	Neoantigen and CpGODN	i.v.	Hepatocellular carcinoma	Bypassed endo-/lysosome degradation, increased antigen uptake and presentation. Increased T cell immunity, inhibition of tumor growth and increased survival	Pre-clinical (in vivo study)	[175]
**PLGA-NP**	hgp10025e33 and TRP2180e188	i.d.	Melanoma	Increased T cell responses and decreased tumor growth	Pre-clinical (in vivo study)	[193]
**Kras peptide vaccine**	KRAS-specific antigens and avasimibe	i.p and i.g.	Lung cancer	Decreased Treg cells and increased cytotoxic T cell tumor infiltration	Pre-clinical (in vivo study)	[195]
**Cationic liposomes**	TAA encoding mRNA	i.v. and i.d.	Prostate cancer	Increase T cell response	Pre-clinical (in vivo study)	[198]
**Liposomes**	MART1 mRNA	i.v.	B16 melanoma	Cellular immune response and induction of anti-tumor cytokines	Pre-clinical (in vivo study)	[199,200]
**Mannosylated NPs- Liposomes**	EPGF and MART1 mRNA	i.v.	B16F10 melanoma	Increased DC activity and anti-tumor immune response	Pre-clinical (in vivo study)	[201]
**Cationic liposomes**	HIV 1 mRNA	i.t.	HIV induced cancer	Increased T cell responses and anti-cancer cytokines	Pre-clinical (in vivo study)	[202]
**Liposomes**	Ovalbumin	Nasal	Melanoma	Increased cytotoxic T cell activity	Pre-clinical (in vivo study)	[203]
**Au-NPs**	Ovalbumin	i.v.	B16 melanoma	Increased anti-tumor activity and survival	Pre-clinical (in vivo study)	[204,205,206]
**Antigen-loaded NPs**	Ovalbumin			Increased DC activity	Pre-clinical (in vitro study)	[207]
**Aluminum hydroxide nanoparticles**	Ovalbumin	i.v.	B16 melanoma	Increased antigen-antibody recognition	Pre-clinical (in vivo study)	[208]
**Chitosan NPs**	Ovalbumin and FITC-BSA	Nasal	B16 melanoma	Increased uptake and presentation of antigen to APCs	Pre-clinical (in vivo study)	[209]
**-γ-PGA NPs**	Ovalbumin	i.d.	B16 melanoma	Helper T cell and cytotoxic T cell response increased	Pre-clinical (in vivo study)	[210]
**Linear polyethylenimine NPs**	MIP3α DNA	i.m.	B-cell non-Hodgkin’s lymphoma	Enhanced Humoral and T cell immune responses	Phase 1	[211]

Abbreviations: i.m: Intramuscular: FITC-BSA: Fluorescein isothiocyanate-labelled bovine serum albumin; PLGA-NP: poly(D, L-lactide-co-glycolide) nanoparticles.

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
