# Peer review of "Advances in Nanotechnology for Cancer Immunoprevention and Immunotherapy: A Review"

_vaccines, 2022, doi:10.3390/vaccines10101727_

Round 1
Reviewer 1 Report
This review is an attempt to connect nanotechnology with the immunoprevention of cancer. While this is a very difficult subject to review, the authors have gone beyond that topic as they try to describe immunotherapy as a broad field but only do so superficially. I would suggest a more focused approach with a detailed description of the different nanoparticles that have been tested in cancer therapy, the biological basis for the adjuvants and descriptions of clinical trials, either completed or currently being conducted. There is also extensive 'English editing needed both in sentence structure and incorrect wording (e.g. line 166 vaccinations contain antigens, not produce antigens).
Author Response
Response to Reviewer 1 Comments
Point 1: This review is an attempt to connect nanotechnology with the immunoprevention of cancer. While this is a very difficult subject to review, the authors have gone beyond that topic as they try to describe immunotherapy as a broad field but only do so superficially. I would suggest a more focused approach with a detailed description of the different nanoparticles that have been tested in cancer therapy, the biological basis for the adjuvants and descriptions of clinical trials, either completed or currently being conducted. There is also extensive 'English editing needed both in sentence structure and incorrect wording (e.g. line 166 vaccinations contain antigens, not produce antigens).
Response 1: Thank you for these critical suggestions. We have modified the title of the manuscript to “Advances in Nanotechnology for Cancer Immunoprevention and Immunotherapy: A Review” that better suits the content of the manuscript. We have tried to adapt a foccused approach and added information on different nanoparticles that have been tested in cancer therapy and description of clinical trial status (pre-clinical, plase I or Phase II) in the manuscript.
We have added more information about applications of various nanoparticles in cancer immunotherapy and immunoprevention in section 6, Nanotechnology in Cancer immunotherapy (lines 248-292) and section 9, Applications of Nanotechnology in and Cancer Vaccines (lines 628-667). We have represented the various examples of nanoparticles used in immunotherapy, immunoprevention and cancer vaccines in Table 2, Table 3 and Table 4. Table 2 enumerates the examples of nanoparticles like gold nanoparticle, liposomes, polymeric NPs that have been extensively used in treatment of cancers like, melanoma, breast cancer, etc. and status of their clinical trials. Nanoparticles that are effective in immunoprevention strategies are enumerated in table 3 titled “Applications of Nanotechnology in Cancer Immunoprevention”, along with the clinical trial status of each approach. Further, we modified Table 4 to include information on status of clinical trials for nanoparticles specifically used in development of cancer vaccines for immunotherapy and immunoprevention of cancer.
Finally, We have thoroughly checked the manuscript for incorrect sentense structure and wording, and have made changes wherever necessary.
Corrections:
- Nanotechnology in Cancer immunotherapy
To overcome the limitations of conventional cancer immunotherapies, nanoparticles have been intensively explored in the area of drug delivery due to their capacity to carry medications to target regions effectively, shield pharmaceuticals from proteolytic enzyme, and stay in circulation for prolonged periods of time [47](Figure 2). For cancer immunotherapy to be successful, three things are crucial. To begin, it is essential for cancer antigens to be successfully transmitted to immune cells, particularly APCs. When cancer antigens and an adjuvant are administered to immune cells, the adjuvant must stimulate an anti-cancer immune response. Thirdly, the immunosuppressive tumor microenvironment (TME) must be regulated for the anti-cancer immunotherapeutic to be effective. Clinical results with cancer vac-cinations have been disappointing thus far. To overcome the limitations of conven-tional cancer immunotherapies, nanoparticles have been intensively explored in the area of drug delivery due to their capacity to carry medications to target regions ef-fectively, shield pharmaceuticals from proteolytic enzymes, and stay in circulation for prolonged periods of time [45](Figure 2). Recent developments in nanotechnology have made it possible to load many components, including tiny molecules, peptides, nucleic acids, and cell membranes, onto structures like liposomes, polymer nanoparti-cles, and inorganic nanoparticles. This makes it possible to co-load antigen and adju-vant in nano-vaccines, ensuring that these active components are administered at the same time to the same APC. Additionally, nano-vaccines promote the effective ac-cumulation of components, such as adjuvant and antigen, in draining lymph nodes and delay their fast spread into circulation [66,67]. Nanoparticle-based vaccinations might thus be useful weapons for boosting the immune system and preventing tumor spread [66]. For instance, Song et al. created a nanoplatform for the delivery of an adjuvant and an antigen by covering PLGA nanoparticles with phospholipid mem-branes. The substantial decrease in the number of metastatic nodules shows that this nano-vaccine may effectively concentrate in lymph nodes and elicit an anti-gen-specific adaptive T cell response, which reduced the metastasis of B16-OVA mel-anoma cells [68].
As a consequence of the potential prospects of nanotechnology and immuno-therapy for treating cancer metastasis, several inventive and intelligent nanomateri-als, including nanorobots, have been developed to enhance therapeutic efficacy. In 2018, Li and colleagues developed a DNA nanorobot that use DNA origami to deliver payloads precisely to tumors. These nanorobots were able to serve as molecularly sen-sitive, precise drug delivery systems that delivered thrombin to blood vessels in solid tumors, resulting in intravascular thrombosis and, ultimately, tumor death [69]. DNA origami scaffolds produced by complementary base pairing provided an advanced drug delivery technology that precisely regulated the number and placement of func-tional moieties, which in turn altered drug loading and stimulus-responsive behavior. Only recently, Li and colleagues [70] developed a DNA-based cancer vaccine that was effectively delivered to the lymph nodes that drain tumors and provided tumor anti-gens to APCs to induce antitumor immune responses. The vaccine contained two types of molecular adjuvants, and an antigen peptide were put together using a tubu-lar DNA nanostructure. Antigens and adjuvants that were previously imprisoned were made visible as a result of the pH-responsive DNA origami being freed within acidic endosomes. These antigens and adjuvants subsequently attached to their re-ceptors, causing DC activation and antigen presentation, which resulted in T cell ac-tivation and cancer cell cytotoxicity. The DNA nanodevice vaccination elicited a strong, tumor-specific T cell immune response that subsequently caused the tumors in mice to shrink as well as an extended T cell immunological memory response that markedly protected animals from tumor metastasis [70,71].
The use of nanomedicines has exciting prospects for boosting the efficiency of such vaccinations. Different nanoplatforms like Immunoliposome, gold nano-particle, iron oxide, PLGA nanocarrier have been studied for their potential to de-livery to enhance anti-tumor immunity and decrease unwanted side effects by transporting molecular, cellular, or subcellular vaccines to lymphoid tissues and cells [45,72].[47,68] Table 2 enumerates examples of various such nanoparticles explored in immunotherapy of cancers. The different applications of nanotechnology in immu-notherapy include nanoparticles used to deliver tumor antigen, adjuvants and TME immunomodulators.
Table 2. Advances of Nanotechnology in Immunotherapy
Nanoparticle |
Active agent |
Delivery method |
Cancer type |
Effect/Inference |
Clinical Trial Status |
Reference |
CNT-CpG |
CpG ODN |
i.tm. |
Subcutaneous Melanomas |
Eradicated glioma and increased tumor immunity |
Pre-clinical (in vivo study) |
[80] |
CNT |
Tumor lysate |
|
Human NSCLC |
Promoted lymphocyte mediated cytotoxicity by NF- ΚB |
Pre-clinical (in vitro study) |
[81] |
HS-TEX |
Chemokines (CCL2, CCL3, CCL4, CCL5, and CCL20) |
i.tm. |
Lung and skin cancer |
Increase activation of T cell and dendritic cells |
Pre-clinical (in vivo study) |
[82] |
AuNPs |
CpG ODN |
i.tm. |
B16 melanoma |
Promoted macrophage and dendritic cell invasion into tumor, inhibited tumor growth and increased survival. |
Pre-clinical (in vivo studies) |
[83,84] |
Hyaluronic acid |
CpG ODN |
i.tm. |
Lymphoma |
Enhanced antitumor activity and immune memory |
Pre-clinical (in vivo study) |
[85] |
Iron Oxide NPs |
CpG ODN |
i.p. |
Colon cancer |
Increased t cell responses and decreased tumor growth |
Pre-clinical (in vivo study) |
[86] |
Liposomes |
Trp2 peptide |
i.v. |
B16 melanoma and lung metastasis |
Enhance T cell responses |
Pre-clinical (in vivo study) |
[87] |
Polymeric NPs (PC7A NP) |
Ovalbumin |
i.v. |
Melanoma, lung, and colon tumor |
Improve delivery of tumor antigen, increase surface presentation and inhibit tumor growth |
Pre-clinical (in vivo study) |
[88] |
Oligonucleotide Nanoring |
Anti-Bmi1 and anti-Mel 18 shRNA with CpG ODN |
i.tm. |
Medulloblastoma |
Inhibit tumor proliferation and growth |
Pre-clinical (in vivo study) |
[89] |
Liposomes |
E7 peptide |
s.c. |
Lung cancer |
Activate antigen presenting cells and stimulate DCs |
Pre-clinical (in vivo study) |
[77] |
R8-Lip |
α-galactosylceramide |
i.v. |
Lung cancer and malignant B16 melanoma |
Activate NK cells and increased anti-tumor immune reesponse |
Pre-clinical (in vivo study) |
[90] |
PLGA-NPs |
TRP2180-188 and 7-acyl lipid A |
s.c. |
B16 Melanoma |
Induces interferon secretion, activates of T cell responses, and decreases tumor size. |
Pre-clinical (in vivo study) |
[91] |
Polymeric NPs |
CpG ODN |
i.d. |
B16 Melanoma |
Activate DCs and inhibited tumor growth |
Pre-clinical (in vivo study) |
[92] |
Protein cage NPs |
Ovalbumin |
i.v. |
B16 Melanoma |
Activate cytotoxic T cells and suppressed tumor growth |
Pre-clinical (in vivo study) |
[93] |
Cowpea mosaic virus nanoparticles |
|
i.t. |
Melanoma, colon, breast, lung and ovarian cancer |
Prevents lung melanoma and generated anti-tumor immunity |
Pre-clinical (in vivo study) |
[94] |
CHP nanogel |
Truncated 146HER2 protein |
s.c. |
HER2 expressing tumor patients |
Induced HER2-specific humoral responses in patients with HER2-expressing tumors |
Phase I |
[95] |
Liposomes |
RNA encoding tumor antigens |
i.v. |
Melanoma |
Induce effector and memory T cell responses, cause INF-α release from macrophages, |
Phase I |
[96] |
Virus-like NPs (MelQbG10) |
Melan-A/MART-1 Peptides with Montanide and Imiquimod |
i.ln |
Melanoma (Stage III-IV) |
Enhanced memory and effector CD8+ T-cell responses |
Phase IIa
|
[78] |
Virus-like NPs |
Melan-A/MART-1 Peptides |
i.d |
Melanoma (Stage II-IV) |
Increased antigen presentation to DC cells and enhanced T cell responses |
Phase IIa |
[97] |
Exosomes |
MAGE 3 peptides |
i.d. |
Melanoma (Stage III-IV) |
Promote tumor rejection and increase T cell responses |
Phase II |
[98] |
Table 3: Applications of Nanotechnology in Cancer Immunoprevention
Nanoparticle |
Active agent |
Delivery method |
Cancer type |
Effect/Inference |
Clinical Trial Status |
Reference |
Iron oxide beads |
Ovalbumin |
s.c. |
B16 Melanoma |
Induced CD8 dependent protective immunity in vivo |
Pre-clinical (in vivo study) |
[181] |
Polystyrene microspheres |
Ovalbumin |
s.c. |
T cell Lymphoma |
Protected against tumor growth and treated existing tumors |
Pre-clinical (in vivo study) |
[189] |
LPH-NPs |
TGF-β si-RNA |
i.v. |
Melanoma |
Knockdown of TGF-β and inhibited tumor growth by 52%. Increased activity of cytotoxic T cell and decreased level of T regs cells |
Pre-clinical (in vivo study) |
[183] |
Iron oxide-zinc oxide NPs |
CEA |
i.v. |
colon adenocarcinoma |
Enhanced T cell responses, reduced tumor growth and better survival |
Pre-clinical (in vivo study) |
[86] |
γ-PGA NPs |
Ovalbumin |
Nasal |
|
Induced antigen specific cellular and humoral immunity |
Pre-clinical (in vivo study) |
[190] |
Liposomes |
CpG-ODN |
i.m. |
B-cell lymphoma |
Induce strong cellular and humoral immunity |
Pre-clinical (in vivo study) |
[184] |
Cationic liposomes |
CpG |
i.d. |
Melanoma |
Increased DC maturation |
Pre-clinical (in vivo study) |
[185] |
Liposomal polymeric gels |
Cyclodextrins, TGF-β inhibitor and IL-2 |
i.tm. |
Melanoma |
Delayed tumor growth and increased tumor survival |
Pre-clinical (in vivo study) |
[186] |
Cationic liposomes |
TLR agonist (CpG ODN) and Ovalbumin |
s.c. or i.d. |
Melanoma |
Increased antigen presentation and enhanced T cell responses |
Pre-clinical (in vivo study) |
[187] |
Cationic liposomes |
α-GalCer with CpG and Ovalbumin |
s.c. |
B16 Melanoma |
Increased activation of NK, DC and T cells |
Pre-clinical (in vivo study) |
[188] |
Tumor cell membrane coated PLGA NPs |
Ovalbumin and PAM or CpG |
|
Melanoma |
Increased antigen presentation and immune responses |
Pre-clinical (in vitro study) |
[191] |
Tumor cell membrane coated NPs |
HLA-Ig and anti-CD28 |
|
Melanoma |
Promoted tumor specific immune response and induced antigen specific activation of T cell |
Pre-clinical (in vitro study) |
[192] |
Latex beads |
Trp2 peptide and CpG |
s.c. and i.v. |
Melanoma |
Inhibited tumor growth and enhanced T cell responses |
Pre-clinical (in vivo study) |
[193] |
iron-dextran particles and quantum dot nanocrystals |
HLA-Ig and anti-CD28 |
i.p and i.v |
Melanoma |
Generation of antigen specific cytotoxic T lymphocytes |
Pre-clinical (in vivo study) |
[182] |
aAPCs |
Trp- 2 peptide |
i.v. |
Melanoma and lung metastasis |
Enhance T cell responses and reduces tumor growth |
Pre-clinical (in vivo study) |
[194] |
- Advanced Applications of Nanotechnology in and Cancer Vaccines
It is established that nanotechnology has been applied to various strategies of immunotherapy and immunoprevention. It has several advantages like it has im-proved the access to the lymph nodes, better tracking, and improved antigen presen-tation, further leading to increased anti-tumor immune response. Immunoprevention seeks to the progression of cancer, and research is being conducted to assess the via-bility of utilizing the theoretical underpinnings of immunoprevention for cancer types that are not linked with infectious agents [198]. Although immunomodulation, anti-bodies are also emerging cancer prevention strategies that are being investigated, prophylactic cancer vaccines are the most effective cancer preventive strategy [199]
The cancer vaccines work in delivery of district components like antigens, ad-juvants or antigen presenting cells. They could either act like an adjuvant themselves or elicit an immune enhancing ability. Various components like mRNA, subunits, peptides, DNA, neoantigen and even whole cells are used as antigens in the prepara-tion of cancer vaccines [161][200]. Table 4 enumerates a number of different nano-particles that have successfully induced and increased anti-cancer immune responses in various cancer types like melanoma, lymphoma, breast cancer, colon cancer etc. Further manyvarious properties of nanoparticles are also crucial to be considered for best outcomes. Properties like particle size, rigidity, surface charge, targeting ligand, and finally immunomodulatory agent added all contribute to the efficiency of the nanoparticle used in the development of cancer vaccines [161][200].
Numerous types of nanoparticles are being used in development of distinct can-cer vaccines that targets some specific part of immune system response and elicit an anti-cancer response. Liposomes are sone such category of nanoparticles (NPs), that have been demonstrated to easily pass- through lipid membrane of various immune cells and cause their activation [162–165][201–204]. Nanoparticles made of inorganic materials like gold and aluminiumaluminum are other class of nanoparticles that are quite popular due to their nontoxic and immunologically inert nature. [166–169][205–208]. Polymeric NPs made up of Chitosan, PGLA etcetc. are also frequently used in cancer vaccines production [157,170][195,209]. These nanoparticles can contain var-ious active compounds that could impart the prophylactic or therapeutic effect.
9.1 Nanotechnology in peptide-based vaccines
Tumor eradication necessitates the production of MHC I-restricted cytotoxic T lym-phocytes (CTLs). This is accomplished by delivering TAA as a peptide or gene in con-junction with strong activation of DCs, which can then stimulate TAA-specific T cells. Trp2, has been identified as a melanoma TAA and has been tailored to various nano-platforms [87,210]. Xu et al.[87] devised a polyplex preparation by varying the ratios of arginine-modified Trp2 and CpG. Furthermore, co-encapsulation of Trp2 peptide and CpG co-within lipid calcium phosphate nanoparticle (LCP NPs) leads to efficient delivery into DCs thereby reducing the tumor burden [87]. DCs are known to better phagocytose cationic nanoparticles (CNPs) when compared to other cell types. The antigen-presenting ability together with the immunostimulatory properties of DCs efficiently initiates T cell responses, and triggers rapid uptakes of CNPs, thereby boosting the immunogenicity of cancer vaccines [211,212]. PLGA polymeric nanopar-ticles are indeed encouraging TAA delivery platforms. This is essential when ad-dressing TLR7/8 agonist delivery, including peptide/protein based TAAs, which are generally confined by limited retention at the administered region. PLGA nanoparti-cles are a desirable delivery platform for such TLR agonists as they proficient-ly enter endosomes/lysosomes upon cytosolic delivery [213].
9.2 Nanotechnology in nucleic acid-based vaccines
Only a small percentage of patients experienced mild therapeutic effects when using peptide-based antigens as cancer vaccines [214]. The combination of genomic se-quencing and nanotechnology has enabled the creation of effective, reliable, and per-sonalized DNA or mRNA vaccines against specific TAA [215]. Nonviral pDNA or mRNA vaccines delivered via nanocarriers are safer and more cost effective than tra-ditional vaccines. This idea was successfully proved with lipidoid nanoparticles [214]. The fact that pDNA/mRNA vaccines elicit both CTLs and helper T cells simultane-ously via both MHC class I and II pathways is a significant benefit [216,217]. Nu-merous polymer and lipid platforms were employed to complex with pDNA for ther-apeutic vaccine applications to enable expression. Chitosan [218], PLL [215], and PEI [219] are examples.
9.3 Nanotechnology in tumor cell or lysate-based vaccines
Applications of tumor cell components like membranes in the development of nano-particle-based cancer vaccines is gaining increased attention due to their ability to mimic the characteristics of the tumor cells. Polymeric nanoparticle when coated with layer of membrane coating derived from tumor cells, presented a plethora of tumor antigens and promoted tumor specific immune response [190]. Another study utilized artificial antigen presenting cells that are coated with human leukocyte antigen–immunoglobulin fusion protein (HLA-Ig) and CD28-specific antibody. These particles were able to activate tumor specific immune response in melanoma cell lines. Further, T cell responses were also enhanced [191].
(Modified) Table 4. Advanced of Nanotechnology in Cancer Vaccines for Immunotherapy and Immuno-prevention.
Nanoparticle |
Active agent |
Delivery method |
Cancer type |
Effect/Inference |
Clinical Trial Status |
Reference |
Au-NPs |
Mangiferin |
i.v. |
Prostate cancer |
Enhanced levels of anti-tumor cytokines with reduced pro-tumor cytokines |
Pre-clinical (in vivo study) |
[221] |
GDNPs 2 |
Ginger bioactive constituents |
Oral and i.p. |
Colitis-Associated Cancer |
Control immune response and chronic inflammation |
Pre-clinical (in vivo study) |
[180] |
Se-NPs-enriched Probiotic |
Lactobacillus plantarum strain |
Oral and i.v. |
Breast cancer murine |
Levels of proinflammatory cytokines increased and increased NK cell activity. Decreased tumor volume and increased survival |
Pre-clinical (in vivo study) |
[195] |
Thiolated nano-vaccine |
Neoantigen and CpGODN |
i.v. |
Hepatocellular carcinoma |
Bypass endo-/lysosome degradation, increased antigen uptake and presentation. Increased T cell immunity, inhibition of tumor growth and increased survival |
Pre-clinical (in vivo study) |
[178] |
PLGA-NP |
hgp10025e33 and TRP2180e188 |
i.d. |
Melanoma |
Increased T cell responses and decreased tumor growth |
Pre-clinical (in vivo study) |
[196] |
Kras peptide vaccine |
KRAS-specific antigens and avasimibe |
i.p and i.g. |
Lung cancer |
Decrease Tregs cell and increased cytotoxic T cell tumor infiltration |
Pre-clinical (in vivo study) |
[198] |
Cationic liposomes |
TAA encoding mRNA |
i.v. and i.d.. |
Prostate cancer |
Increase T cell response |
Pre-clinical (in vivo study) |
[222] |
Liposomes |
MART1 mRNA |
i.v. |
B16 melanoma |
Cellular immune response and induction of anti-tumor cytokines |
Pre-clinical (in vivo study) |
[202,203] |
Mannosylated NPs- Liposomes |
EPGF and MART1 mRNA |
i.v. |
B16F10 melanoma |
Increased DC activity and anti-tumor immune response |
Pre-clinical (in vivo study) |
[205] |
Cationic liposomes |
HIV 1 mRNA |
i.t. |
HIV induced cancer |
Increased T cell responses and anti-cancer cytokines |
Pre-clinical (in vivo study) |
[223] |
Liposomes |
Ovalbumin |
Nasal |
Melanoma |
Increased cytotoxic T cell activity |
Pre-clinical (in vivo study) |
[204] |
Au-NPs |
Ovalbumin |
i.v. |
B16 melanoma |
Increased anti-tumor activity and survival |
Pre-clinical (in vivo study) |
[207,208,224] |
Antigen-loaded NPs |
Ovalbumin |
|
|
Increased DC activity |
Pre-clinical (in vitro study) |
[225] |
Aluminum hydroxide nanoparticles |
Ovalbumin |
i.v. |
B16 melanoma |
Increased antigen-antibody recognition |
Pre-clinical (in vivo study) |
[209] |
Chitosan NPs |
Ovalbumin and FITC-BSA |
Nasal |
B16 melanoma |
Increased uptake and presentation of antigen to APCs |
Pre-clinical (in vivo study) |
[210] |
-γ-PGA NPs |
Ovalbumin |
i.d. |
B16 melanoma |
Helper T cell and cytotoxic T cell response increase |
Pre-clinical (in vivo study) |
[226] |
Linear polyethylenimine NPs |
MIP3α DNA |
i.m. |
B-cell non-Hodgkin’s lymphoma |
Enhanced Humoral and T cell immune responses |
Phase 1 |
[218] |
Reviewer 2 Report
The manuscript entitled 'Advances in nanotechnology for cancer immunoprevention: A review' addresses important questions regarding cancer therapy and prevention. However, it is my opinion that structural alterations must be performed for the manuscript to become acceptable for publication:
Major revisions:
1. The title is somewhat misleading: the manuscript has much information about immunotherapy and immunoprevention but information regarding the use of nanotechnology is short, taking in account that this is the major topic of the title. The authors should either change the title or increase the information about nanotechnology utilization for cancer treatment prevention. It would greatly enhance the interest of the manuscript the addition of information of vaccines or other type of therapies (already in use or in clinical trials) that currently use nanotechnology. Table 2 should be improved with information regarding the status of the technology (i.e., already approved for clinical use, in clinical trials, in vivo studies, in vitro studies...)
2. An index should be added to the manuscript. It is very difficult to follow the section division of the manuscript. When reading, it is very confusing and hard to understand if we following for a different section or if t is a sub-section. This needs to be improved.
3. Cancer is a group of diseases that are very diverse between each other, thus is very difficult to find one treatment suitable for all types of cancer. Although the authors refer the different types of cancers when giving examples of prevention strategies, in some parts of the text it seems that cancer is being addressed in a generic approach which does not reflect the diversity of cancer. The authors should take this in account throughout the text and discuss the application of nanotechnology taking in mind the specific type of cancer, the availability of treatment/cure and the degree of cancer severity.
Minor revision:
A thorough proof reading of the manuscript should be conducted to correct small mistakes. Here are a few that I have detected:
line14 - immunothe apies
line89 - text format in table1
line283 - TGF-
line334 - ofvaccines
line364 - arecaused
line371 - present nboth
line451 - asspciated
Author Response
Response to Reviewer 2 Comments
The manuscript entitled 'Advances in nanotechnology for cancer immunoprevention: A review' addresses important questions regarding cancer therapy and prevention. However, it is my opinion that structural alterations must be performed for the manuscript to become acceptable for publication:
Major revisions:
Point 1: The title is somewhat misleading: the manuscript has much information about immunotherapy and immunoprevention but information regarding the use of nanotechnology is short, taking in account that this is the major topic of the title. The authors should either change the title or increase the information about nanotechnology utilization for cancer treatment prevention. It would greatly enhance the interest of the manuscript the addition of information of vaccines or other type of therapies (already in use or in clinical trials) that currently use nanotechnology. Table 2 should be improved with information regarding the status of the technology (i.e., already approved for clinical use, in clinical trials, in vivo studies, in vitro studies...)
Response 1: We thank the reviewer for their thoughtful and valuable suggestions. Following their comment, we have now added more information regarding applications of nanotechnology in cancer immunotherapy and immunoprevention. We have also changed the title of the manuscript to “Advances in Nanotechnology for Cancer Immunoprevention and Immunotherapy: A Review” As suggested by the reviewer we have included tables (Table 2 and Table 3) that enumerate the applications of nanoparticles in cancer immunotherapy and immunoprevention along with the status of the clinical trials. Table 4 (Modified Table 2) is also improoved and contains information regarding the status of the technology.
Corrections:
- Nanotechnology in Cancer immunotherapy
To overcome the limitations of conventional cancer immunotherapies, nanoparticles have been intensively explored in the area of drug delivery due to their capacity to carry medications to target regions effectively, shield pharmaceuticals from proteolytic enzyme, and stay in circulation for prolonged periods of time [47](Figure 2). For cancer immunotherapy to be successful, three things are crucial. To begin, it is essential for cancer antigens to be successfully transmitted to immune cells, particularly APCs. When cancer antigens and an adjuvant are administered to immune cells, the adjuvant must stimulate an anti-cancer immune response. Thirdly, the immunosuppressive tumor microenvironment (TME) must be regulated for the anti-cancer immunotherapeutic to be effective. Clinical results with cancer vac-cinations have been disappointing thus far. To overcome the limitations of conven-tional cancer immunotherapies, nanoparticles have been intensively explored in the area of drug delivery due to their capacity to carry medications to target regions ef-fectively, shield pharmaceuticals from proteolytic enzymes, and stay in circulation for prolonged periods of time [45](Figure 2). Recent developments in nanotechnology have made it possible to load many components, including tiny molecules, peptides, nucleic acids, and cell membranes, onto structures like liposomes, polymer nanoparti-cles, and inorganic nanoparticles. This makes it possible to co-load antigen and adju-vant in nano-vaccines, ensuring that these active components are administered at the same time to the same APC. Additionally, nano-vaccines promote the effective ac-cumulation of components, such as adjuvant and antigen, in draining lymph nodes and delay their fast spread into circulation [66,67]. Nanoparticle-based vaccinations might thus be useful weapons for boosting the immune system and preventing tumor spread [66]. For instance, Song et al. created a nanoplatform for the delivery of an adjuvant and an antigen by covering PLGA nanoparticles with phospholipid mem-branes. The substantial decrease in the number of metastatic nodules shows that this nano-vaccine may effectively concentrate in lymph nodes and elicit an anti-gen-specific adaptive T cell response, which reduced the metastasis of B16-OVA mel-anoma cells [68].
As a consequence of the potential prospects of nanotechnology and immuno-therapy for treating cancer metastasis, several inventive and intelligent nanomateri-als, including nanorobots, have been developed to enhance therapeutic efficacy. In 2018, Li and colleagues developed a DNA nanorobot that use DNA origami to deliver payloads precisely to tumors. These nanorobots were able to serve as molecularly sen-sitive, precise drug delivery systems that delivered thrombin to blood vessels in solid tumors, resulting in intravascular thrombosis and, ultimately, tumor death [69]. DNA origami scaffolds produced by complementary base pairing provided an advanced drug delivery technology that precisely regulated the number and placement of func-tional moieties, which in turn altered drug loading and stimulus-responsive behavior. Only recently, Li and colleagues [70] developed a DNA-based cancer vaccine that was effectively delivered to the lymph nodes that drain tumors and provided tumor anti-gens to APCs to induce antitumor immune responses. The vaccine contained two types of molecular adjuvants, and an antigen peptide were put together using a tubu-lar DNA nanostructure. Antigens and adjuvants that were previously imprisoned were made visible as a result of the pH-responsive DNA origami being freed within acidic endosomes. These antigens and adjuvants subsequently attached to their re-ceptors, causing DC activation and antigen presentation, which resulted in T cell ac-tivation and cancer cell cytotoxicity. The DNA nanodevice vaccination elicited a strong, tumor-specific T cell immune response that subsequently caused the tumors in mice to shrink as well as an extended T cell immunological memory response that markedly protected animals from tumor metastasis [70,71].
The use of nanomedicines has exciting prospects for boosting the efficiency of such vaccinations. Different nanoplatforms like Immunoliposome, gold nano-particle, iron oxide, PLGA nanocarrier have been studied for their potential to de-livery to enhance anti-tumor immunity and decrease unwanted side effects by transporting molecular, cellular, or subcellular vaccines to lymphoid tissues and cells [45,72].[47,68] Table 2 enumerates examples of various such nanoparticles explored in immunotherapy of cancers. The different applications of nanotechnology in immu-notherapy include nanoparticles used to deliver tumor antigen, adjuvants and TME immunomodulators.
Table 2. Advances of Nanotechnology in Immunotherapy
Nanoparticle |
Active agent |
Delivery method |
Cancer type |
Effect/Inference |
Clinical Trial Status |
Reference |
CNT-CpG |
CpG ODN |
i.tm. |
Subcutaneous Melanomas |
Eradicated glioma and increased tumor immunity |
Pre-clinical (in vivo study) |
[80] |
CNT |
Tumor lysate |
|
Human NSCLC |
Promoted lymphocyte mediated cytotoxicity by NF- ΚB |
Pre-clinical (in vitro study) |
[81] |
HS-TEX |
Chemokines (CCL2, CCL3, CCL4, CCL5, and CCL20) |
i.tm. |
Lung and skin cancer |
Increase activation of T cell and dendritic cells |
Pre-clinical (in vivo study) |
[82] |
AuNPs |
CpG ODN |
i.tm. |
B16 melanoma |
Promoted macrophage and dendritic cell invasion into tumor, inhibited tumor growth and increased survival. |
Pre-clinical (in vivo studies) |
[83,84] |
Hyaluronic acid |
CpG ODN |
i.tm. |
Lymphoma |
Enhanced antitumor activity and immune memory |
Pre-clinical (in vivo study) |
[85] |
Iron Oxide NPs |
CpG ODN |
i.p. |
Colon cancer |
Increased t cell responses and decreased tumor growth |
Pre-clinical (in vivo study) |
[86] |
Liposomes |
Trp2 peptide |
i.v. |
B16 melanoma and lung metastasis |
Enhance T cell responses |
Pre-clinical (in vivo study) |
[87] |
Polymeric NPs (PC7A NP) |
Ovalbumin |
i.v. |
Melanoma, lung, and colon tumor |
Improve delivery of tumor antigen, increase surface presentation and inhibit tumor growth |
Pre-clinical (in vivo study) |
[88] |
Oligonucleotide Nanoring |
Anti-Bmi1 and anti-Mel 18 shRNA with CpG ODN |
i.tm. |
Medulloblastoma |
Inhibit tumor proliferation and growth |
Pre-clinical (in vivo study) |
[89] |
Liposomes |
E7 peptide |
s.c. |
Lung cancer |
Activate antigen presenting cells and stimulate DCs |
Pre-clinical (in vivo study) |
[77] |
R8-Lip |
α-galactosylceramide |
i.v. |
Lung cancer and malignant B16 melanoma |
Activate NK cells and increased anti-tumor immune reesponse |
Pre-clinical (in vivo study) |
[90] |
PLGA-NPs |
TRP2180-188 and 7-acyl lipid A |
s.c. |
B16 Melanoma |
Induces interferon secretion, activates of T cell responses, and decreases tumor size. |
Pre-clinical (in vivo study) |
[91] |
Polymeric NPs |
CpG ODN |
i.d. |
B16 Melanoma |
Activate DCs and inhibited tumor growth |
Pre-clinical (in vivo study) |
[92] |
Protein cage NPs |
Ovalbumin |
i.v. |
B16 Melanoma |
Activate cytotoxic T cells and suppressed tumor growth |
Pre-clinical (in vivo study) |
[93] |
Cowpea mosaic virus nanoparticles |
|
i.t. |
Melanoma, colon, breast, lung and ovarian cancer |
Prevents lung melanoma and generated anti-tumor immunity |
Pre-clinical (in vivo study) |
[94] |
CHP nanogel |
Truncated 146HER2 protein |
s.c. |
HER2 expressing tumor patients |
Induced HER2-specific humoral responses in patients with HER2-expressing tumors |
Phase I |
[95] |
Liposomes |
RNA encoding tumor antigens |
i.v. |
Melanoma |
Induce effector and memory T cell responses, cause INF-α release from macrophages, |
Phase I |
[96] |
Virus-like NPs (MelQbG10) |
Melan-A/MART-1 Peptides with Montanide and Imiquimod |
i.ln |
Melanoma (Stage III-IV) |
Enhanced memory and effector CD8+ T-cell responses |
Phase IIa
|
[78] |
Virus-like NPs |
Melan-A/MART-1 Peptides |
i.d |
Melanoma (Stage II-IV) |
Increased antigen presentation to DC cells and enhanced T cell responses |
Phase IIa |
[97] |
Exosomes |
MAGE 3 peptides |
i.d. |
Melanoma (Stage III-IV) |
Promote tumor rejection and increase T cell responses |
Phase II |
[98] |
Table 3: Applications of Nanotechnology in Cancer Immunoprevention
Nanoparticle |
Active agent |
Delivery method |
Cancer type |
Effect/Inference |
Clinical Trial Status |
Reference |
Iron oxide beads |
Ovalbumin |
s.c. |
B16 Melanoma |
Induced CD8 dependent protective immunity in vivo |
Pre-clinical (in vivo study) |
[181] |
Polystyrene microspheres |
Ovalbumin |
s.c. |
T cell Lymphoma |
Protected against tumor growth and treated existing tumors |
Pre-clinical (in vivo study) |
[189] |
LPH-NPs |
TGF-β si-RNA |
i.v. |
Melanoma |
Knockdown of TGF-β and inhibited tumor growth by 52%. Increased activity of cytotoxic T cell and decreased level of T regs cells |
Pre-clinical (in vivo study) |
[183] |
Iron oxide-zinc oxide NPs |
CEA |
i.v. |
colon adenocarcinoma |
Enhanced T cell responses, reduced tumor growth and better survival |
Pre-clinical (in vivo study) |
[86] |
γ-PGA NPs |
Ovalbumin |
Nasal |
|
Induced antigen specific cellular and humoral immunity |
Pre-clinical (in vivo study) |
[190] |
Liposomes |
CpG-ODN |
i.m. |
B-cell lymphoma |
Induce strong cellular and humoral immunity |
Pre-clinical (in vivo study) |
[184] |
Cationic liposomes |
CpG |
i.d. |
Melanoma |
Increased DC maturation |
Pre-clinical (in vivo study) |
[185] |
Liposomal polymeric gels |
Cyclodextrins, TGF-β inhibitor and IL-2 |
i.tm. |
Melanoma |
Delayed tumor growth and increased tumor survival |
Pre-clinical (in vivo study) |
[186] |
Cationic liposomes |
TLR agonist (CpG ODN) and Ovalbumin |
s.c. or i.d. |
Melanoma |
Increased antigen presentation and enhanced T cell responses |
Pre-clinical (in vivo study) |
[187] |
Cationic liposomes |
α-GalCer with CpG and Ovalbumin |
s.c. |
B16 Melanoma |
Increased activation of NK, DC and T cells |
Pre-clinical (in vivo study) |
[188] |
Tumor cell membrane coated PLGA NPs |
Ovalbumin and PAM or CpG |
|
Melanoma |
Increased antigen presentation and immune responses |
Pre-clinical (in vitro study) |
[191] |
Tumor cell membrane coated NPs |
HLA-Ig and anti-CD28 |
|
Melanoma |
Promoted tumor specific immune response and induced antigen specific activation of T cell |
Pre-clinical (in vitro study) |
[192] |
Latex beads |
Trp2 peptide and CpG |
s.c. and i.v. |
Melanoma |
Inhibited tumor growth and enhanced T cell responses |
Pre-clinical (in vivo study) |
[193] |
iron-dextran particles and quantum dot nanocrystals |
HLA-Ig and anti-CD28 |
i.p and i.v |
Melanoma |
Generation of antigen specific cytotoxic T lymphocytes |
Pre-clinical (in vivo study) |
[182] |
aAPCs |
Trp- 2 peptide |
i.v. |
Melanoma and lung metastasis |
Enhance T cell responses and reduces tumor growth |
Pre-clinical (in vivo study) |
[194] |
- Advanced Applications of Nanotechnology in and Cancer Vaccines
It is established that nanotechnology has been applied to various strategies of immunotherapy and immunoprevention. It has several advantages like it has im-proved the access to the lymph nodes, better tracking, and improved antigen presen-tation, further leading to increased anti-tumor immune response. Immunoprevention seeks to the progression of cancer, and research is being conducted to assess the via-bility of utilizing the theoretical underpinnings of immunoprevention for cancer types that are not linked with infectious agents [198]. Although immunomodulation, anti-bodies are also emerging cancer prevention strategies that are being investigated, prophylactic cancer vaccines are the most effective cancer preventive strategy [199]
The cancer vaccines work in delivery of district components like antigens, ad-juvants or antigen presenting cells. They could either act like an adjuvant themselves or elicit an immune enhancing ability. Various components like mRNA, subunits, peptides, DNA, neoantigen and even whole cells are used as antigens in the prepara-tion of cancer vaccines [161][200]. Table 4 enumerates a number of different nano-particles that have successfully induced and increased anti-cancer immune responses in various cancer types like melanoma, lymphoma, breast cancer, colon cancer etc. Further manyvarious properties of nanoparticles are also crucial to be considered for best outcomes. Properties like particle size, rigidity, surface charge, targeting ligand, and finally immunomodulatory agent added all contribute to the efficiency of the nanoparticle used in the development of cancer vaccines [161][200].
Numerous types of nanoparticles are being used in development of distinct can-cer vaccines that targets some specific part of immune system response and elicit an anti-cancer response. Liposomes are sone such category of nanoparticles (NPs), that have been demonstrated to easily pass- through lipid membrane of various immune cells and cause their activation [162–165][201–204]. Nanoparticles made of inorganic materials like gold and aluminiumaluminum are other class of nanoparticles that are quite popular due to their nontoxic and immunologically inert nature. [166–169][205–208]. Polymeric NPs made up of Chitosan, PGLA etcetc. are also frequently used in cancer vaccines production [157,170][195,209]. These nanoparticles can contain var-ious active compounds that could impart the prophylactic or therapeutic effect.
9.1 Nanotechnology in peptide-based vaccines
Tumor eradication necessitates the production of MHC I-restricted cytotoxic T lym-phocytes (CTLs). This is accomplished by delivering TAA as a peptide or gene in con-junction with strong activation of DCs, which can then stimulate TAA-specific T cells. Trp2, has been identified as a melanoma TAA and has been tailored to various nano-platforms [87,210]. Xu et al.[87] devised a polyplex preparation by varying the ratios of arginine-modified Trp2 and CpG. Furthermore, co-encapsulation of Trp2 peptide and CpG co-within lipid calcium phosphate nanoparticle (LCP NPs) leads to efficient delivery into DCs thereby reducing the tumor burden [87]. DCs are known to better phagocytose cationic nanoparticles (CNPs) when compared to other cell types. The antigen-presenting ability together with the immunostimulatory properties of DCs efficiently initiates T cell responses, and triggers rapid uptakes of CNPs, thereby boosting the immunogenicity of cancer vaccines [211,212]. PLGA polymeric nanopar-ticles are indeed encouraging TAA delivery platforms. This is essential when ad-dressing TLR7/8 agonist delivery, including peptide/protein based TAAs, which are generally confined by limited retention at the administered region. PLGA nanoparti-cles are a desirable delivery platform for such TLR agonists as they proficient-ly enter endosomes/lysosomes upon cytosolic delivery [213].
9.2 Nanotechnology in nucleic acid-based vaccines
Only a small percentage of patients experienced mild therapeutic effects when using peptide-based antigens as cancer vaccines [214]. The combination of genomic se-quencing and nanotechnology has enabled the creation of effective, reliable, and per-sonalized DNA or mRNA vaccines against specific TAA [215]. Nonviral pDNA or mRNA vaccines delivered via nanocarriers are safer and more cost effective than tra-ditional vaccines. This idea was successfully proved with lipidoid nanoparticles [214]. The fact that pDNA/mRNA vaccines elicit both CTLs and helper T cells simultane-ously via both MHC class I and II pathways is a significant benefit [216,217]. Nu-merous polymer and lipid platforms were employed to complex with pDNA for ther-apeutic vaccine applications to enable expression. Chitosan [218], PLL [215], and PEI [219] are examples.
9.3 Nanotechnology in tumor cell or lysate-based vaccines
Applications of tumor cell components like membranes in the development of nano-particle-based cancer vaccines is gaining increased attention due to their ability to mimic the characteristics of the tumor cells. Polymeric nanoparticle when coated with layer of membrane coating derived from tumor cells, presented a plethora of tumor antigens and promoted tumor specific immune response [190]. Another study utilized artificial antigen presenting cells that are coated with human leukocyte antigen–immunoglobulin fusion protein (HLA-Ig) and CD28-specific antibody. These particles were able to activate tumor specific immune response in melanoma cell lines. Further, T cell responses were also enhanced [191].
(Modified) Table 4. Advanced of Nanotechnology in Cancer Vaccines for Immunotherapy and Immuno-prevention.
Nanoparticle |
Active agent |
Delivery method |
Cancer type |
Effect/Inference |
Clinical Trial Status |
Reference |
Au-NPs |
Mangiferin |
i.v. |
Prostate cancer |
Enhanced levels of anti-tumor cytokines with reduced pro-tumor cytokines |
Pre-clinical (in vivo study) |
[221] |
GDNPs 2 |
Ginger bioactive constituents |
Oral and i.p. |
Colitis-Associated Cancer |
Control immune response and chronic inflammation |
Pre-clinical (in vivo study) |
[180] |
Se-NPs-enriched Probiotic |
Lactobacillus plantarum strain |
Oral and i.v. |
Breast cancer murine |
Levels of proinflammatory cytokines increased and increased NK cell activity. Decreased tumor volume and increased survival |
Pre-clinical (in vivo study) |
[195] |
Thiolated nano-vaccine |
Neoantigen and CpGODN |
i.v. |
Hepatocellular carcinoma |
Bypass endo-/lysosome degradation, increased antigen uptake and presentation. Increased T cell immunity, inhibition of tumor growth and increased survival |
Pre-clinical (in vivo study) |
[178] |
PLGA-NP |
hgp10025e33 and TRP2180e188 |
i.d. |
Melanoma |
Increased T cell responses and decreased tumor growth |
Pre-clinical (in vivo study) |
[196] |
Kras peptide vaccine |
KRAS-specific antigens and avasimibe |
i.p and i.g. |
Lung cancer |
Decrease Tregs cell and increased cytotoxic T cell tumor infiltration |
Pre-clinical (in vivo study) |
[198] |
Cationic liposomes |
TAA encoding mRNA |
i.v. and i.d.. |
Prostate cancer |
Increase T cell response |
Pre-clinical (in vivo study) |
[222] |
Liposomes |
MART1 mRNA |
i.v. |
B16 melanoma |
Cellular immune response and induction of anti-tumor cytokines |
Pre-clinical (in vivo study) |
[202,203] |
Mannosylated NPs- Liposomes |
EPGF and MART1 mRNA |
i.v. |
B16F10 melanoma |
Increased DC activity and anti-tumor immune response |
Pre-clinical (in vivo study) |
[205] |
Cationic liposomes |
HIV 1 mRNA |
i.t. |
HIV induced cancer |
Increased T cell responses and anti-cancer cytokines |
Pre-clinical (in vivo study) |
[223] |
Liposomes |
Ovalbumin |
Nasal |
Melanoma |
Increased cytotoxic T cell activity |
Pre-clinical (in vivo study) |
[204] |
Au-NPs |
Ovalbumin |
i.v. |
B16 melanoma |
Increased anti-tumor activity and survival |
Pre-clinical (in vivo study) |
[207,208,224] |
Antigen-loaded NPs |
Ovalbumin |
|
|
Increased DC activity |
Pre-clinical (in vitro study) |
[225] |
Aluminum hydroxide nanoparticles |
Ovalbumin |
i.v. |
B16 melanoma |
Increased antigen-antibody recognition |
Pre-clinical (in vivo study) |
[209] |
Chitosan NPs |
Ovalbumin and FITC-BSA |
Nasal |
B16 melanoma |
Increased uptake and presentation of antigen to APCs |
Pre-clinical (in vivo study) |
[210] |
-γ-PGA NPs |
Ovalbumin |
i.d. |
B16 melanoma |
Helper T cell and cytotoxic T cell response increase |
Pre-clinical (in vivo study) |
[226] |
Linear polyethylenimine NPs |
MIP3α DNA |
i.m. |
B-cell non-Hodgkin’s lymphoma |
Enhanced Humoral and T cell immune responses |
Phase 1 |
[218] |
Point 2: An index should be added to the manuscript. It is very difficult to follow the section division of the manuscript. When reading, it is very confusing and hard to understand if we following for a different section or if t is a sub-section. This needs to be improved.
Response 2: Kindly accept our sincere thanks for providing this important remark. As suggested by the reviewer we have now added the index in the manuscript.
Point 3: Cancer is a group of diseases that are very diverse between each other, thus is very difficult to find one treatment suitable for all types of cancer. Although the authors refer the different types of cancers when giving examples of prevention strategies, in some parts of the text it seems that cancer is being addressed in a generic approach which does not reflect the diversity of cancer. The authors should take this in account throughout the text and discuss the application of nanotechnology taking in mind the specific type of cancer, the availability of treatment/cure and the degree of cancer severity.
Response 3: We thank the reviewer for pointing out this ambiguity and we apologize for being unclear. We have modified the information in the revised manuscript to answer this question. We have discussed the applications of nanotechnolgy with respect to the specific types of cancers.
Minor revision:
Point 4: A thorough proof reading of the manuscript should be conducted to correct small mistakes. Here are a few that I have detected:
line14 - immunothe apies
line89 - text format in table1
line283 - TGF-
line334 - ofvaccines
line364 - arecaused
line371 - present nboth
line451 - asspciated
Response: We thank the reviewer for their thoughtful and informative feedback and we apologize for the inadvertent errors. We have thoroughly checked the manuscript for incorrect sentense structure and wording, and have made changes wherever necessary.
Round 2
Reviewer 1 Report
no further comments
Author Response
We thank the reviewer for their valuable suggestions and believe that the quality of the manuscript has significantly increased due to their feedback.
Reviewer 2 Report
The manuscript file provided is not formatted adequately for proper revision. There are still issues with formatting that make it difficult to follow the text. Please provide a new version with all the alterations for correct revision. Also, although the authors response mention an index, in the manuscript the index is still missing.
Round 3
Reviewer 2 Report
Dear authors, please add the index in the beginning of the manuscript. I accept the manuscript for publication and I have no further comments to make.